# Sleep homeostasis regulated by 5HT2b receptor in a small subset of neurons in the dorsal fan-shaped body of drosophila

**Yongjun Qian[1,2], Yue Cao[1], Bowen Deng[1,2], Guang Yang[1], Jiayun Li[1†], Rui Xu[3], Dandan zhang[3], Juan Huang[3]\*, Yi Rao[1,2]\***

[1]Peking-Tsinghua Center for Life Sciences, State Key Laboratory of Biomembrane and Membrane Biology, PKU-IDG/McGovern Institute For Brain Research, Beijing Advanced Innovation Center for Genomics, School of Life Sciences, Peking University, Beijing, China; [2]National Institute of Biological Sciences, Beijing, China; [3]School of Basic Medical Sciences, Nanjing Medical University, Nanjing, China

**Abstract** Our understanding of the molecular mechanisms underlying sleep homeostasis is limited. We have taken a systematic approach to study neural signaling by the transmitter 5-hydroxytryptamine (5-HT) in drosophila. We have generated knockout and knockin lines for *Trh*, the 5-HT synthesizing enzyme and all five 5-HT receptors, making it possible for us to determine their expression patterns and to investigate their functional roles. Loss of the *Trh*, *5HT1a* or *5HT2b* gene decreased sleep time whereas loss of the *Trh* or *5HT2b* gene diminished sleep rebound after sleep deprivation. *5HT2b* expression in a small subset of, probably a single pair of, neurons in the dorsal fan-shaped body (dFB) is functionally essential: elimination of the *5HT2b* gene from these neurons led to loss of sleep homeostasis. Genetic ablation of 5HT2b neurons in the dFB decreased sleep and impaired sleep homeostasis. Our results have shown that serotonergic signaling in specific neurons is required for the regulation of sleep homeostasis.
DOI: https://doi.org/10.7554/eLife.26519.001

**\*For correspondence:**
huangjuan@njmu.edu.cn (JH);
yrao@pku.edu.cn (YR)

**Present address:** †Cold Spring Harbor Laboratory, Cold Spring Harbor, United States

**Competing interests:** The authors declare that no competing interests exist.

## Introduction

Most of us spend nearly a third of life in sleep, though the mechanisms underlying sleep remain unclear. Circadian rhythm and homeostasis can separately and interactively regulate sleep and wakefulness (*Borbély, 1982*; *Borbély and Achermann, 1999*). Research in drosophila and other organisms has revealed the molecular mechanisms of the circadian clock (reviewed in *Panda et al. [2002]*). By contrast, less is known about the molecular mechanisms controlling sleep homeostasis.

The development of drosophila as a model animal to study sleep (*Hendricks et al., 2000*; *Shaw et al., 2000*) has led to findings of multiple genes that are important for sleep (*Afonso et al., 2015*; *Bushey et al., 2009*; *Chung et al., 2009*; *Cirelli et al., 2005*; *Crocker and Sehgal, 2008*; *Crocker et al., 2010*; *Donlea et al., 2014*; *Foltenyi et al., 2007*; *Guo et al., 2011*; *Koh et al., 2008*; *Kume et al., 2005*; *Li et al., 2013*; *Liu et al., 2008, 2014*; *Metaxakis et al., 2014*; *Naik et al., 2008*; *Parisky et al., 2008*; *Park et al., 2014*; *Shimizu et al., 2008*; *Soshnev et al., 2011*; *Stavropoulos and Young, 2011*; *Takahama et al., 2012*; *Tomita et al., 2011*; *Ueno et al., 2012*; *Yuan et al., 2006*), whereas less is known about sleep homeostasis, operationally defined as sleep rebound after deprivation. Molecular components that are implicated in sleep homeostasis include cyclic AMP, *CREB*, *sleepless*, *cyc*, *Hsp83*, *Cullin-3* and *cyclin A* (*Cirelli et al., 2005*; *Foltenyi et al., 2007*; *Koh et al., 2008*; *Rogulja and Young, 2012*; *Shaw et al., 2002*; *Stavropoulos and Young, 2011*; *Vanderheyden et al., 2013*). The lLNv and DN1 clock neurons are important for circadian control of sleep (*Agosto et al., 2008*; *Chung et al., 2009*; *Guo et al., 2016*;

*Liu et al., 2014*; *Parisky et al., 2008*) while the ellipsoid body (EB), the mushroom bodies (MB) and the ExFl2 neurons projecting to dorsal fan-shaped body (dFB) are important for sleep homeostasis (*Donlea et al., 2014*; *Liu et al., 2016*; *Pimentel et al., 2016*; *Sitaraman et al., 2015*). dFB neurons alter their excitability in response to sleep deprivation, which is mediated by a specific Rho-GTPase-activating protein (Rho-Gap), *crossveinless-c* (*Donlea et al., 2014*). R2 neurons in the EB act upstream of dFB and specifically generate sleep drive (*Liu et al., 2016*). Neurotransmitters are known to be involved in regulating sleep in mammals (*Nall and Sehgal, 2014*). In flies, thermogenetic activation of a small subset of cholinergic neurons promotes sleep and elicits sleep homeostasis, whereas activation of octopaminergic neurons promotes sleep but suppresses sleep homeostasis (*Seidner et al., 2015*).

Serotonin or 5-hydroxytryptamine (5-HT) is involved in multiple behaviors in drosophila, including learning and memory, feeding, courtship and aggression (*Becnel et al., 2011*; *Dierick, 2007*; *Johnson et al., 2011*; *Liu et al., 2011*; *Sitaraman et al., 2008*; *Yuan et al., 2006*). The *5HT1a* receptor is important for sleep: *5HT1a* mutant flies had reduced and fragmented sleep (*Yuan et al., 2006*). However, the involvement of 5-HT in sleep homeostasis is unclear.

## Results

### Investigation of the serotonergic system in drosophila

5-HT is synthesized in two steps: the conversion of tryptophan to 5-hydroxytryptophan (5HTP) by tryptophan hydroxylase (Trh in flies and Tph in mammals) (*Kuhn et al., 1979*), followed by the conversion of 5HTP to 5-HT by aromatic amino acid decarboxylase (*Figure 1A*). We have generated genetic tools that allow systematic studies of the serotonergic system. Four receptors were known to be 5-HT receptors in drosophila when we started this project: *5HT1a, 5HT1b, 5HT2* and *5HT7* (*Colas et al., 1995*; *Saudou et al., 1992*; *Witz et al., 1990*). Our bioinformatics analysis had revealed a new G-protein- coupled receptor (GPCR), annotated as *CG42796* in the drosophila genome, as a new 5-HT receptor which we named as *5HT2b*, with the previously known *5HT2* becoming *5HT2a* (*Brody and Cravchik, 2000*; *Gasque et al., 2013*). 5HT2b was predicted to be coupled to Gq protein and shown in our collaboration to mediate 5-HT responsiveness (*Gasque et al., 2013*).

We constructed knockout and knockin lines for Trh and all the five receptors for 5HT in drosophila. The ends-out gene targeting strategy was used to generate null mutants (*Huang et al., 2009*; *Rong and Golic, 2000*) (Strategy I) (*Figure 1B*, *Figure 1—figure supplement 1A and B*). To visualize gene expression and to manipulate neuronal activity, we generated *Gal4*, *LexA* or *Flp* knock-in lines for each of these genes using the CRISPR/CAS9 method, with *Gal4*, *LexA* or *Flp* introduced at the starting sequence of the first exon (Strategy II) or at the end of the open reading frame (Strategy III) (*Figure 1C and D*, *Figure 1—figure supplement 1C and D*). Using Strategy II, we obtained null mutants with *Gal4/Flp/LexA* replacing the targeted gene, which provided strains that allowed us to manipulate neurons in mutants lacking specific genes. Strategy III enabled us to visualize the expression of specific genes faithfully and manipulated neurons without interrupting gene function and expression. The genotypes of constructed lines were confirmed by polymerase chain reaction (PCR) and sequencing (*Figure 1—figure supplement 1*).

We also constructed an indel mutant for *Trh* (Trh[01]) with a guide RNA targeting the catalytic center of the enzyme and deleting two basepairs (*Figure 1—figure supplement 1E–F*), which caused an early translational stop (*Figure 1—figure supplement 1G*). No 5-HT was detected in the brains of Trh[01] or Trh[GKO] mutants (generated by Strategy II, *Gal4* insertion and replacement of gene), although it was detected in the brains of the wild type (wt) or Trh::Gal4 (generated by Strategy III) flies (*Figure 1—figure supplement 2A–D*).

### 5-HT regulation of sleep through both the 5HT2b and 5HT1a receptors

We analyzed the sleep patterns of *Trh* and 5-HT receptor mutants (*Figure 1E*). *Trh*, *5HT1a* or *5HT2b* mutant flies were found to sleep less than the wt in both 12 hr (hr) light/12 hr dark (LD) cycles and in constant darkness (DD) (*Figure 1F–H*, *Figure 1—figure supplement 3A–C*). These three mutants exhibited reduced sleep bout duration during the day and night (*Figure 1I*), and also in DD phase (*Figure 1—figure supplement 3D and E*). *Trh*, *5HT1a* and *5HT2b* mutants also displayed prolonged

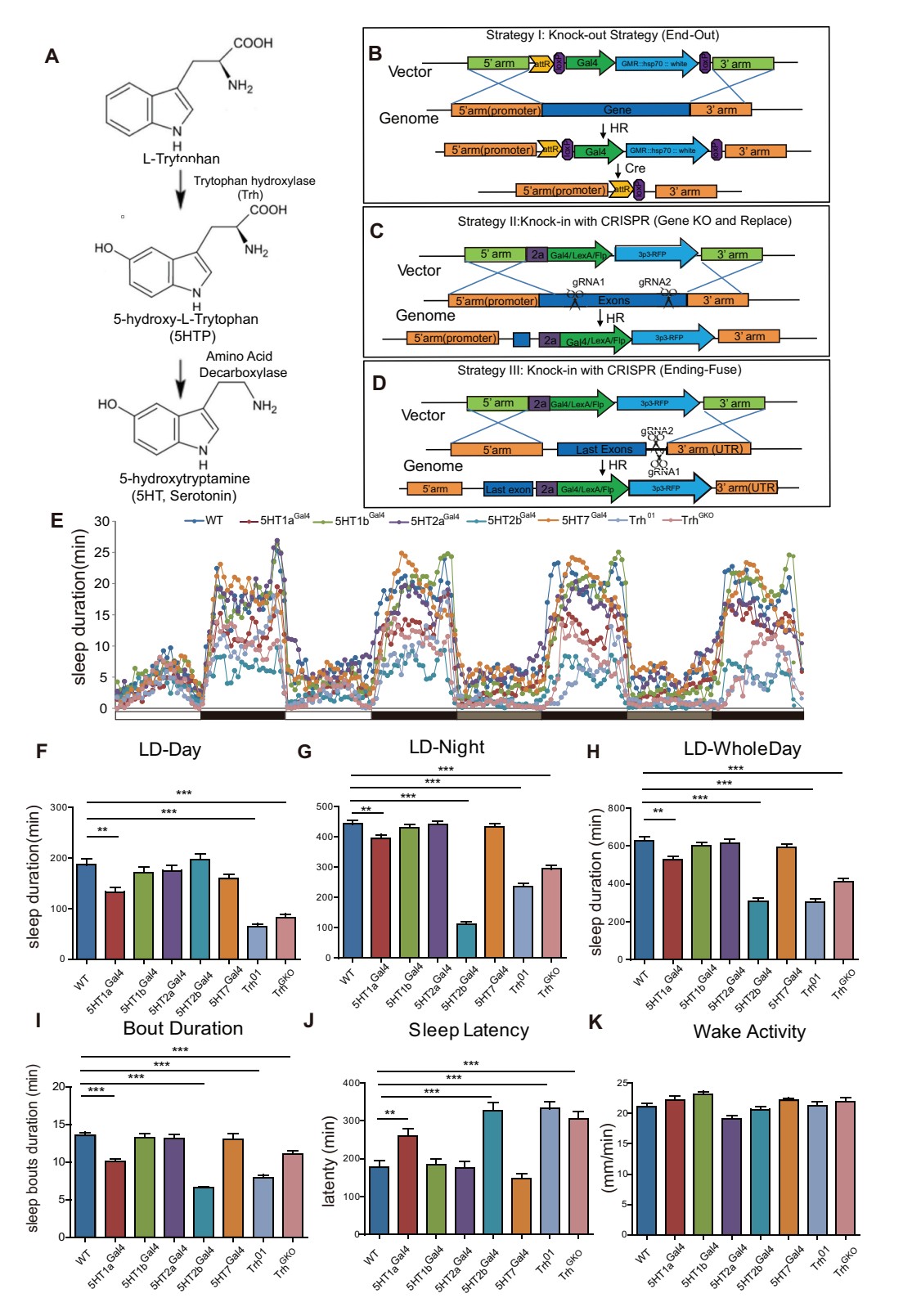

**Figure 1.** Sleep in Trh and 5-HT receptor mutants. (A) 5-HT synthesis in the brain. (B–D) Three strategies used in constructing drosophila lines. (B) The end-out method. All five receptor mutants were constructed by this method. (C and D) The CRISPR/Cas9 system was used to delete target DNA and/or to insert DNA. *Gal4, Flp* or *LexA* were introduced at specific locations, such as the beginning of the first exon (C), or the end of the open reading frame (ORF) with the 2a peptide as linker (D). (E) Sleep profiles over four consecutive days, the first two days were in 12 hr light/12 hr dark (LD) cycle,

*Figure 1 continued on next page*

*Figure 1 continued*
and last two days in constant darkness (DD). (F–H) *Trh*, *5HT1a* and *5HT2b* mutant flies slept less than wild type (wt) controls. Sleep bout durations in LD are shown for: nighttime (F), daytime (G) and whole 24 hour day (H). (I) Sleep bout duration was reduced in Trh[01], Trh[GKO], *5HT1a* and *5HT2b* mutants during LD. (J) Latency after light-off was delayed in Trh[01], Trh[GKO], *5HT1a* and *5HT2b* mutants. (K) All mutants showed normal activity when awake, as measured by locomoter distance per waking min. (B–K, mean ± SEM, n = 48 for WT, n = 37 for *5HT1a* mutants, n = 48 for *5HT1b* mutants, n = 45 for *5HT2a* mutants, n = 40 for *5HT2b* mutants, n = 48 for 5HT7 mutants, n = 43 for Trh[01] mutants, and n = 48 for Trh[GKO] mutant).
DOI: https://doi.org/10.7554/eLife.26519.002
The following figure supplements are available for figure 1:

**Figure supplement 1.** Genotype confirmation for constructed lines.
DOI: https://doi.org/10.7554/eLife.26519.003
**Figure supplement 2.** 5HTP restored serotonin expression and behavioral defects in *Trh* mutant flies.
DOI: https://doi.org/10.7554/eLife.26519.004
**Figure supplement 3.** Sleep phenotype of female flies.
DOI: https://doi.org/10.7554/eLife.26519.005
**Figure supplement 4.** Sleep in *Trh* and 5-HT receptor male mutant flies.
DOI: https://doi.org/10.7554/eLife.26519.006

latency to sleep at night (*Figure 1J*). *Trh*, *5HT1a* and *5HT2b* mutants showed no change in the intensity of locomotor activity when awake (*Figure 1K*). As reported previously, *5HT1a* mutant flies had reduced sleep (*Yuan et al., 2006*). The sleep phenotype of *5HT2b* mutants was much more severe than that of *5HT1a* mutants, nearly equal to that of *Trh* mutants (*Figure 1E and H*). In DD, *5HT2b* mutants showed a large decrease in sleep duration and sleep bout duration in both the subjective daytime and the subjective nighttime (*Figure 1—figure supplement 3A–C*). Results in males were similar to those in females except that male *5HT1b* mutants exhibited decreased sleep during the daytime of LD but not during DD (*Figure 1—figure supplement 4*). These results indicated that *5HT2b* functions to promote sleep. The difference between LD and DD was consistent with the abnormal light sensitivity of *5HT1b* mutants as reported previously (*Yuan et al., 2005*).

To test whether 5-HT functions directly in adult flies or indirectly during development, we restored 5-HT level in *Trh* mutant flies by feeding adults with 5HTP, which could circumvent the requirement for *Trh*. Two mg/ml 5HTP for 3 days restored 5HT immunofluorescence in the brains of Trh[01] and Trh[GKO] flies (*Figure 1—figure supplement 2E–H*). 5-HTP feeding rescued both daytime and nighttime sleep in 5HTP to Trh[01] mutants (*Figure 1—figure supplement 2I and J*) and in Trh[GKO] mutants (data not shown) to about wt level, indicating that 5-HT promotes sleep in adult flies.

## 5-HT regulation of sleep recovery after deprivation through the 5HT2b receptor

The involvement of 5-HT in circadian rhythm has been reported previously (*Nichols, 2007*; *Page, 1987*; *Rea et al., 1994*; *Yuan et al., 2005*). We first analyzed free-running locomotor rhythms to distinguish between the roles of serotonin signaling in circadian and homeostatic sleep regulation. We have not observed circadian period changes in any of our mutants when compared to the wt in the same genetic background after entrainment to a 12 hr (hr) light/12 hr dark (LD) cycle for 3 days, followed by analysis in constant darkness (DD) for 12 days (*Figure 2—figure supplement 1*). Next, we investigated whether 5-HT regulated sleep homeostasis. We deprived flies of sleep for a whole night with approximately 100% efficiency (*Figure 2—figure supplement 2*), and measured sleep recovery over the ensuing 48 hr. Wt flies regained their lost sleep and completely recovered during the first 24 hr (*Figure 2—figure supplement 3C*). *Trh* and *5HT2b* mutants exhibited a significantly lower percentage of sleep recovery rate than wt flies (*Figure 2A and B*). For weaker sleep deprivation, these mutants also showed impaired sleep recovery (*Figure 2—figure supplement 3*). However, *5HT1a* mutant flies, though defective in sleep, were similar to the wt and mutants of any other serotonergic receptors in showing normal sleep recovery after sleep deprivation (*Figure 2A and B*), further supporting the hypothesis that involvement in regulating total sleep time is separate from involvement in regulating sleep recovery after deprivation. To test whether 5-HT functions to control sleep homeostasis in adult flies and not through indirect actions during development, we fed 5-HTP

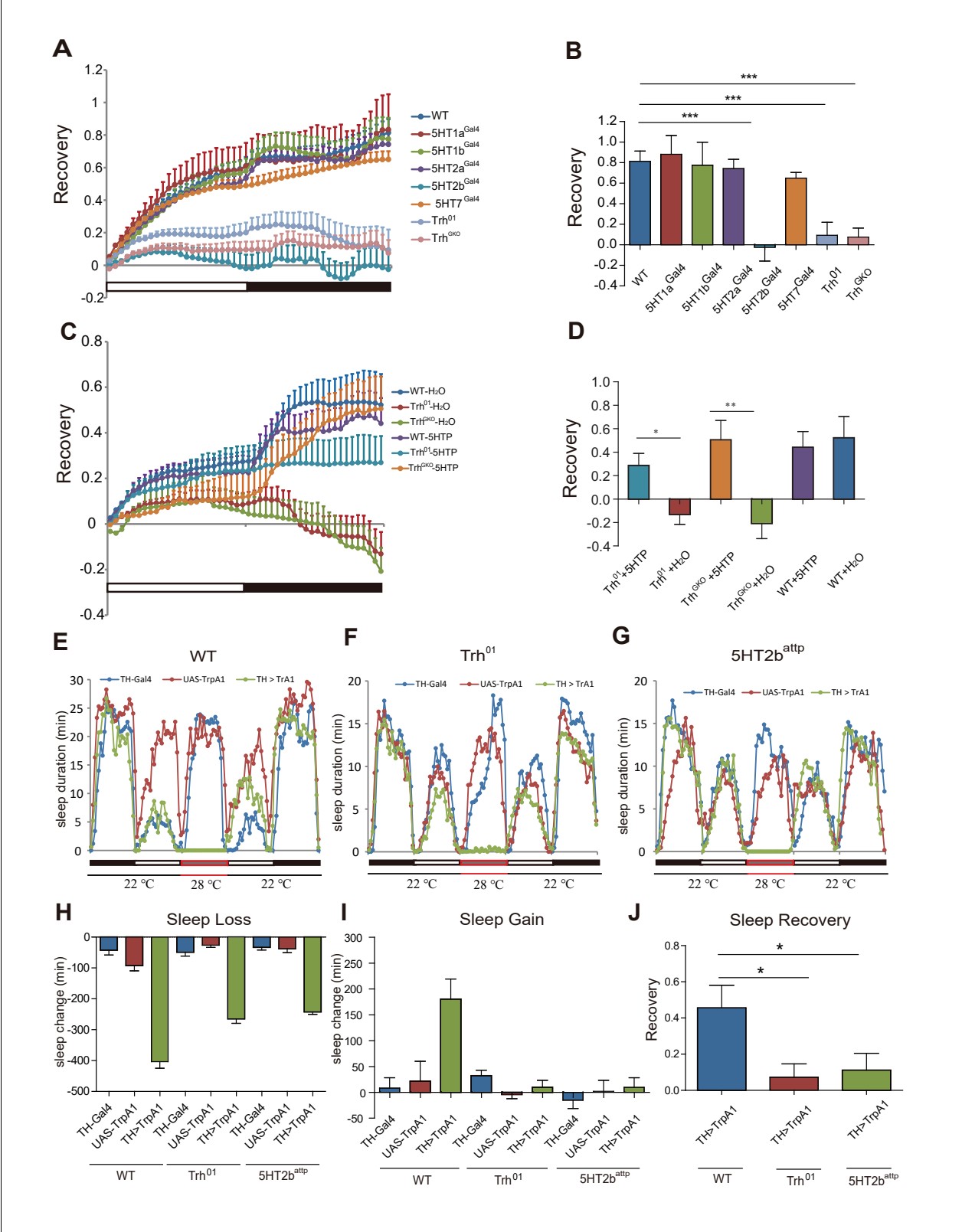

**Figure 2.** Sleep homeostasis in *Trh* and 5-HT receptor mutants. (**A**) *Trh* and *5HT2b* mutant flies regained a lower percentage of lost sleep time than wt after 12-hr overnight sleep deprivation. Other receptor mutants showed normal recovery rate after sleep deprivation. (**B**) Statistical analysis of sleep rebound after 24-hr recovery (mean ± SEM, n = 39 for wt, n = 38 for *5HT1a* mutants, n = 38 for *5HT1b* mutants, n = 38 for *5HT2a* mutants, n = 45 for *5HT2b* mutants, n = 47 for *5HT7* mutants, n = 44 for Trh[01] mutants, and n = 42 for Trh[GKO] mutants, respectively). (**C**) Sleep recovery rate after sleep

*Figure 2 continued on next page*

Figure 2 continued

deprivation. Three-day feeding of 2 mg/ml 5HTP did not change sleep recovery in wt flies, but rescued sleep rebound in Trh[01] and Trh[GKO] flies. (D) Statistical analysis (mean ± SEM, n = 47 for wt with $H_2O$, n = 34 for Trh[01] with $H_2O$, n = 34 for Trh[GKO] with $H_2O$, n = 48 for wt with 5HTP, n = 36 for Trh[01] with 5HTP, and n = 45 for Trh[GKO] with 5HTP). (E–G) Sleep profiles after thermogenetical activation of dopaminergic neurons in wt (E), *Trh* mutant (F), or *5HT2b* mutant flies (G). (H) Sleep loss during neural activation. (I) Sleep gain over 24 hr after thermogenetically induced sleep deprivation. (J) Statistical analysis of sleep rebound after 24-hr recovery. (E–J, n = 32–48 for each strain). One-way ANOVA was used to detect statistical difference between different genotypes. $*p<0.05$, $**p<0.01$, $***p<0.001$.

DOI: https://doi.org/10.7554/eLife.26519.007

The following figure supplements are available for figure 2:

**Figure supplement 1.** Circaidian rhythms in Trh and 5-HT receptor mutant flies.
DOI: https://doi.org/10.7554/eLife.26519.008
**Figure supplement 2.** Efficiency of sleep deprivation.
DOI: https://doi.org/10.7554/eLife.26519.009
**Figure supplement 3.** Sleep homeostasis.
DOI: https://doi.org/10.7554/eLife.26519.010

to adult flies. We found that 5HTP restored sleep homeostasis in *Trh* mutant flies to the normal level (*Figure 2C and D*).

Previous studies have shown that the activation of specific neurons could cause sleep deprivation (*Dubowy et al., 2016*; *Seidner et al., 2015*). We took advantage of this to induce sleep loss thermogenetically by expressing transgenic TrpA1 channels (UAS-TrpA1) in dopaminergic neurons (TH-Gal4). We measured the sleep loss, sleep gain and recovery rate in wt, *Trh* mutant and *5HT2b* mutant flies after sleep loss induced by activation of dopaminergic neurons. *Trh* and *5HT2b* mutant flies showed impaired sleep recovery when compared to wt flies (*Figure 2E–J*). Thus, our results indicate that 5-HT and its 2b receptor regulate sleep homeostasis in adult flies, independent of the methods of sleep deprivation.

## Expression patterns of *Trh* and *5HT2b* genes

To visualize the patterns of *Trh* or *5HT2b*, we crossed flies carrying Trh::Gal4 and 5HT2b::Gal4 (both generated by Strategy III) with flies carrying UAS-mCD8GFP for labeling of the cytoplasmic membrane of cells expressing each gene (*Figure 3A and E*, *Figure 3—figure supplement 1F and J*), UAS-stingerGFP for nuclear labeling (*Figure 3B and F*, *Figure 3—figure supplement 1G and K*), UAS-DscamGFP for dendritic labeling (*Figure 3C and G*, *Figure 3—figure supplement 1H and L*) or UAS-sytGFP for axonal labeling (*Figure 3D and H*, *Figure 3—figure supplement 1I and M*). *Trh*- and *5HT2b*-positive neurons were found in the brain and the ventral nerve cord (VNC). Trh::Gal4 labeled all of the previously reported 5-HT clusters in the adult brain: Anterior lateral protocerebrum (ALP), Anterior medial protocerebrum (AMP), Anterior dorsomedial protocerebrum (ADMP), Lateral protocerebrum (LP), Lateral subesophageal ganglion (SEL) and Medial subesophageal ganglion (SEM) anterior clusters; Posterior lateral protocerebrum (PLP), Posterior medial protocerebrum, dorsal (PMPD), Posterior medial protocerebrum, medial (PMPM) and Posterior medial protocerebrum, ventral (PMPV) posterior clusters (*Alekseyenko et al., 2010*; *Pooryasin and Fiala, 2015*; *Sitaraman et al., 2008 , 2012*; *Vallés and White, 1988*) and showed similar expression pattern when compared to reported transgenic strains (*Alekseyenko et al., 2010*). The distribution of the axons and dendrites of neurons that were positive for Trh::Gal4 could be found in the optic lobe, the olfactory lobe, the central complex and the mushroom bodies (MBs) (*Figure 3C and D*, *Video 1*).

5HT2b::Gal4 labeled more than 500 neurons in the brain (*Figure 3E and F*). The axons and dendrites were found in the central complex, the olfactory lobe, the optic lobe, the subesophageal ganglion and the ventrolateral protocerebrum, but not the MBs (*Figure 3G and H*, *Video 2*). A recent study of *5HT2b* expression using the MiMIC system labeled the same brain regions and showed a similar expression pattern (*Gnerer et al., 2015*). We also examined the expression patterns of Trh[GKO], 5HT2b[GKO] (generated by Strategy II, Gal4 insertion and replacement) and 5HT2b-LexA (generated by Strategy II, LexA insertion and replacement), and found patterns of expression similar to that of the Gal4 line generated by Strategy III (*Figure 3—figure supplement 1A–E*).

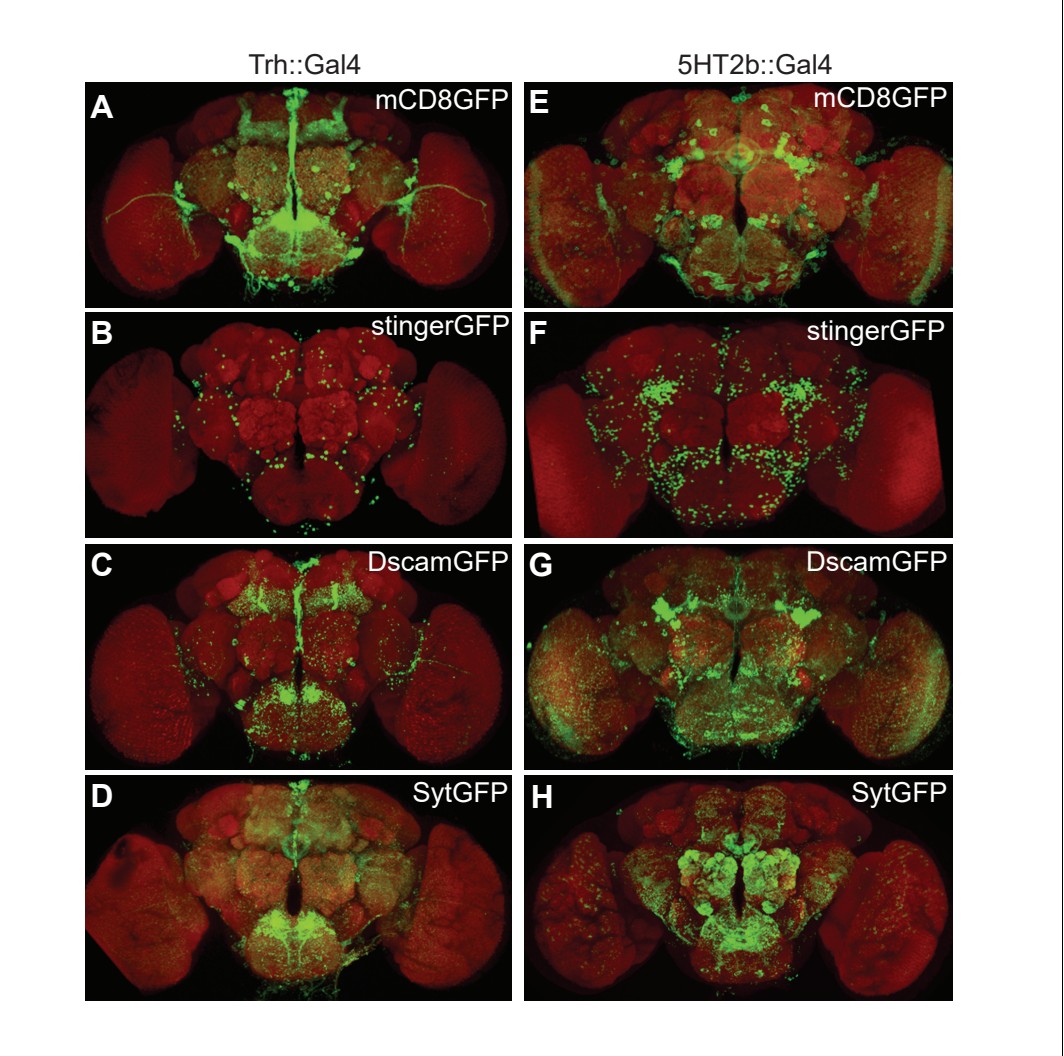

**Figure 3.** Expression patterns of *Trh* and *5HT2b* genes in the brain. (A–D) Brains of UAS-mCD8GFP; Trh::Gal4 (**A**), UAS-stingerGFP; Trh::Gal4 (**B**), UAS-DscamGFP; Trh::Gal4 (**C**) and UAS-sytGFP; Trh::Gal4 (**D**) flies immunostained with the anti-GFP antibody (green) and the neuropil marker nc82 antibody (red). (E–H) Brains of UAS-mCD8GFP; 5HT2b::Gal4 (**E**), UAS-stingerGFP; 5HT2b::Gal4 (**F**), UAS-DscamGFP; 5HT2b::Gal4 (**G**) and UAS-sytGFP; 5HT2b::Gal4 (**H**), immunostained with anti-GFP antibody (green) and nc82 antibody (red).

DOI: https://doi.org/10.7554/eLife.26519.011

The following figure supplement is available for figure 3:

**Figure supplement 1.** Expression patterns of different Trh and 5HT2b strains.

DOI: https://doi.org/10.7554/eLife.26519.012

## Expression of *5HT2b* in a pair of dFB neurons

Previous studies have implicated different brain regions, such as the MB (*Joiner et al., 2006*; *Pitman et al., 2006*; *Sitaraman et al., 2015*), the dFB (*Donlea et al., 2011, 2014*; *Liu et al., 2012*; *Pimentel et al., 2016*; *Ueno et al., 2012*), large ventral lateral clock neurons (lLNv) (*Chung et al., 2009*), dorsal neurons (DN1) (*Guo et al., 2016*; *Kunst et al., 2014*), the ellipsoid body (EB) (*Liu et al., 2016*), and the pars intercerebralis (PI) (*Crocker et al., 2010*) in controlling sleep duration and homeostasis (*Figure 4A*). We used the intersectional strategy to examine whether *5HT2b* was expressed in any of these regions. Sparse expression was found in the MB and extensive expression in the dFB when 5HT2b-LexA strains were combined with LexAop-Flp, UAS-FRT-stop-FRT-mCD8GFP and different region-specific Gal4 lines (*Figure 4—figure supplement 1B*). To further confirm the functional roles of the *5HT2b* gene, we expressed *5HT2b* RNA interference (RNAi) in

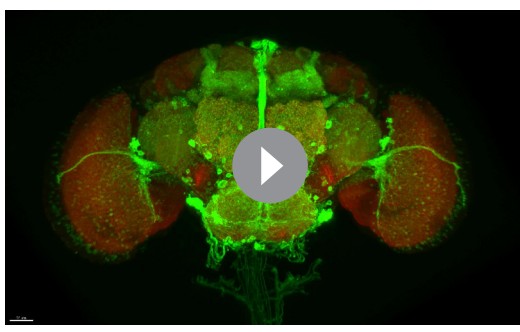

**Video 1.** Expression pattern of the *Trh* gene. About 200 neurons were labeled in the brain. The neural projections were found in optic lobes, olfactory lobes central complex, subesophageal ganglion and mushroom bodies.
DOI: https://doi.org/10.7554/eLife.26519.013

5HT2b neurons and different brain regions. RNAi efficiency was validated by qRT-PCR (*Figure 4—figure supplement 1A*). RNAi-mediated knockdown of either 5HT2b or dFB neurons reduced sleep relative to controls, whereas knockdown of *5HT2b* in other regions had no significant effect (*Figure 4A*). 23E10-GAL4 and 23E10-LexA lines labeled approximately 40 neurons projecting their axons to the dFB (*Figure 4B and C*) (*Donlea et al., 2014*; *Manning et al., 2012*). Two to four dFB neurons were labeled on each side of the adult fly with intersection of 5HT2b and 23E10 (*Figure 4D and E*, *Video 3*). We also used 23E10-Gal4 and 5HT2b-LexA to drive the expression of UAS-StingerGFP and LexAop-tdTomato separately (*Figure 4F*). Only a single dFB neuron on each side of the adult fly was positive for both green fluorescent protein (GFP) and red fluorescent protein (RFP) (*Figure 4G and H*). They constitute a single pair of 5HT2b-positive

neurons in the dFB.

To visualize 5HT2b protein expression, we constructed a 5HT2b protein-trap knockin strain by fusing a superfolder GFP to the carboxyl terminus of 5HT2b protein (*Pédelacq et al., 2006*). We found the pattern of protein expression to be similar to that of *5HT2b* gene expression (*Figure 3E* and *Figure 4J*). We also found both strong anti-serotonin immunofluorescence and 5HT2b$^{sfGFP}$ staining signal in the dorsal FB region (*Figure 4I–K*). To determine whether the ExFl2 neurons contribute to the 5HT2b$^{sfGFP}$ staining signal in the dorsal FB region, we crossed UAS-5HT2b$^{sfGFP}$ to 23E10-Gal4, and found strong GFP in the presynaptic region of ExFl2 neurons, but not in the postsynaptic region nor in the cell body (*Figure 4L–N*). These results suggest that the 5HT2b receptor is located presynaptically in ExFl2 neurons.

## Manipulation of 5HT2b and dFB intersectional neurons regulates sleep duration and homeostasis

Artificial activation of dFB neurons induces sleep (*Donlea et al., 2011*; *Ueno et al., 2012*), whereas silencing of them reduces sleep (*Kottler et al., 2013*; *Liu et al., 2012*). We confirmed these results by activating 23E10-labeled dFB neurons with UAS-Nachbach (*Luan et al., 2006*), which used a bacterial sodium channel to increase neuronal excitability (*Figure 5—figure supplement 1A and B*). We crossed 23E10-Gal4 with UAS-head involution defective (UAS-Hid), expressing a protein causing cell death (*Zhou et al., 1997*), to ablate the 23E10 neurons (*Figure 5—figure supplement 1C and D*). Sleep duration was reduced in flies in which 23E10 neurons were ablated (*Figure 5—figure supplement 1E and F*). Flies without 23E10 neurons lost sleep homeostasis: showing no sleep rebound after 12-hr overnight sleep deprivation (*Figure 5—figure supplement 1G and H*).

We next tested the roles of 5HT2b- and dFB-intersectional neurons. We used Gal80 driven by the *Tublin* promoter to suppress 23E10-Gal4-driven expression of UAS-mCD8GFP, or of effectors such as Nachbach and Hid, with the Gal80 gene flanked by FRT recombination sites (FLP-out Gal80; Tub-FRT-Gal80-FRT). Gal80

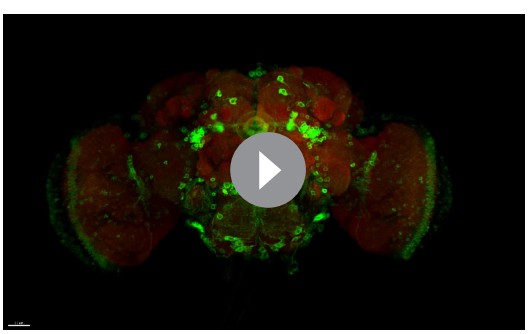

**Video 2.** Expression pattern of the *5HT2b* gene. More than 500 neurons were labeled in the brain. The neural projections were found in the central complex, the olfactory lobe, the optic lobe, the subesophageal ganglion and the ventrolateral protocerebrum, but not in the MBs.
DOI: https://doi.org/10.7554/eLife.26519.014

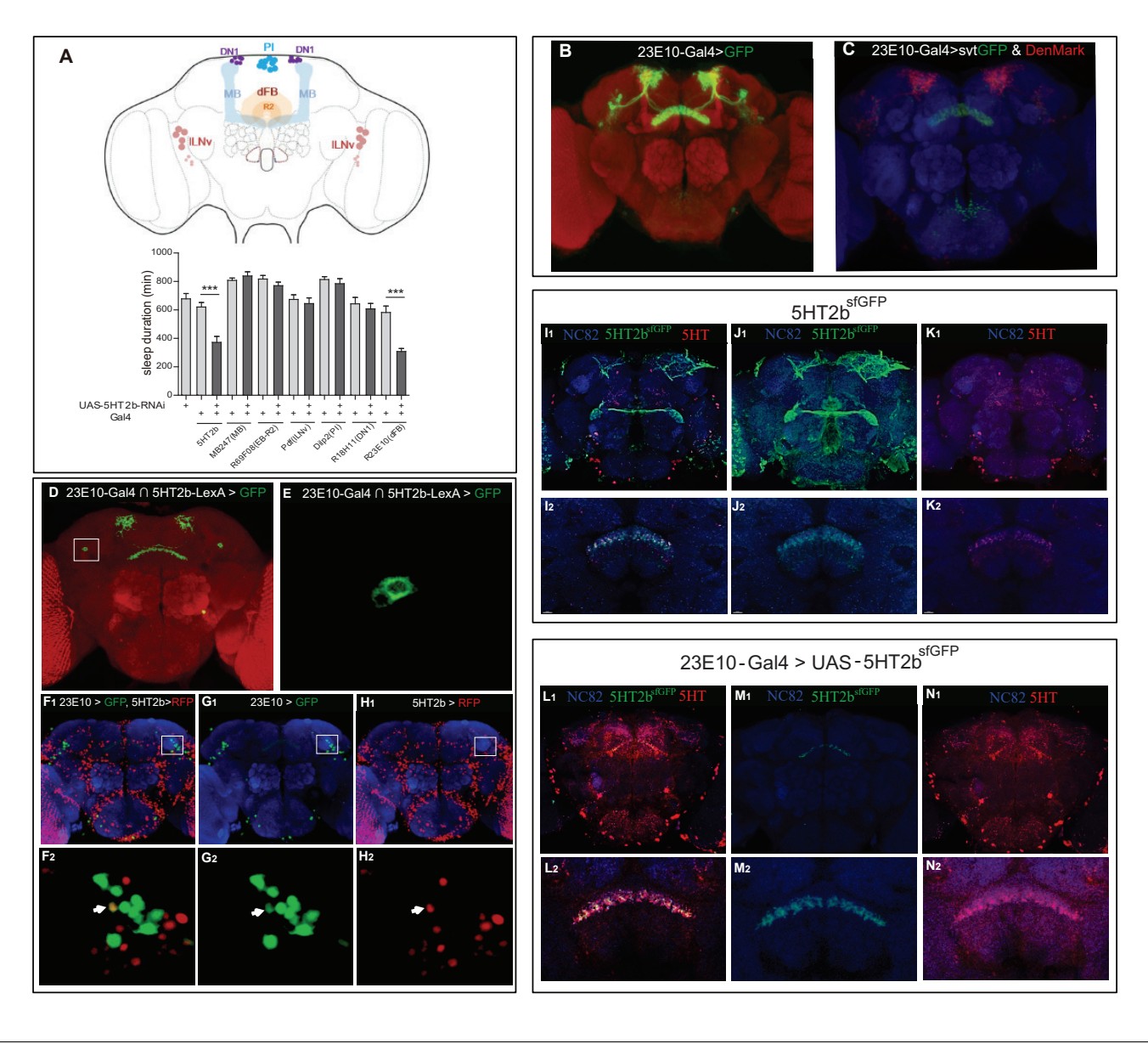

**Figure 4.** A single pair of dFB neurons expressing *5HT2b*. (**A**) Schematic illustration of different brain regions regulating sleep (above). dFB, dorsal fan-shaped body; DN1, dorsal neurons 1; lLNv, large ventral lateral clock neurons; MB, mushroom body; PI, pars intercerebralis. Sleep in flies expressing different Gal4-driven *5HT2b* RNAi lines (below). (**B**) Brain of UAS-mCD8GFP; 23E10-Gal4 flies, immunostained with the anti-GFP antibody (green) and the nc82 antibody (red). (**C**) Brain of UAS-DenMark,UAS-sytGFP; 23E10-Gal4 fly, immunostained with the anti-GFP antibody (green), the anti-RFP antibody (red) and the neuropil marker nc82 antibody (blue). (**D and E**) Intersectional neurons of 23E10 and 5HT2b. Brain of 23E10-LexA/UAS-FRT-mCD8GFP-FRT; LexAop-Flp/Trh::Gal4 fly, immunostained with the anti-GFP antibody (green) and the neuropil marker nc82 antibody (red). Two intersectional neurons were labeled (**E**). (**F–H**) One pair of 5HT2b and 23E10 co-stained neurons. 5HT2b-LexA driven LexAop-tdTomato labelled 5HT2b neurons after anti-RFP immunostaining, and 23E10-Gal4 driven UAS-stingerGFP labelled 23E10 neurons after anti-GFP immunostaining. (F1, G1, and H1) Whole-mount staining. (F2, G2, and H2) Cell bodies of two co-stained neurons. NC82 fluorescences blue. Arrows indicate the cell bodies of co-stained neurons. (**I–K**) Immunostainings of 5HT2b^sfGFP and serotonin. 5HT2b^sfGFP labelled green, 5-HT labelled red and NC82 labelled blue as background. (I1, J1, and K1) Whole-mount staining. (I2, J2, and K2) Slice of FB region staining. (**L–N**) Immunostaining of 5HT2b^sfGFP at dFB and serotonin. UAS-5HT2b^sfGFP driven by 23E10 labelled green, 5HT labelled red and NC82 labelled blue as background. (L1, M1, and N1) Whole-mount staining. (L2, M2, and N2) Slice of FB region.

DOI: https://doi.org/10.7554/eLife.26519.015

The following figure supplement is available for figure 4:

**Figure supplement 1.** Intersections of 5HT2b neurons with different brain regions.

*Figure 4 continued on next page*

*Figure 4 continued*

DOI: https://doi.org/10.7554/eLife.26519.016

could efficiently suppress Gal4 expression (*Figure 6—figure supplement 1C*). When combined 5HT2b-LexA was used to drive the expression of LexAop-Flp, FLP-mediated recombination caused GAL80 to be flipped out, resulting in Gal4 expression in Flp-containing 5HT2b neurons. Consequently, GAL4 was effective exclusively in 5HT2b and 23E10 intersectional neurons (*Figure 6—figure supplement 1A and B*) (*Gordon and Scott, 2009*). Sleep was greatly increased after 23E10 and 5HT2b intersectional neurons were activated (*Figure 5A–C*). Sleep was reduced in flies in which 23E10 and 5HT2b intersectional neurons were ablated (*Figure 5A–C*). Sleep homeostasis was impaired after ablation of 23E10 and 5HT2b intersectional neurons (*Figure 5D–F*).

## Sufficiency of *5HT2b* gene in dFB neurons for sleep homeostasis

To examine whether the 5HT2b receptor functions in dFB neurons, we restored *5HT2b* cDNA to *5HT2b* knockout mutant flies by UAS-5HT2b (*Figure 6A*). 5HT2b$^{GKO}$ is a Gal4 KI line generated by Strategy II which was used to drive the expression of 5HT2b cDNA in all 5HT2b neurons, whereas 23E10 Gal4 was used to drive 5HT2b expression in dFB neurons (*Figure 6A*). Both significantly increased sleep duration at nighttime but neither affected daytime sleep (*Figure 6B and C*), indicating that *5HT2b* in dFB neurons is sufficient to promote nighttime sleep. Sleep recovery after 12-hr deprivation in *5HT2b* mutant flies was rescued by either *5HT2b* expression in all *5HT2b* positive neurons or *5HT2b* expression in 23E10 neurons (*Figure 6D and E*). We also used RNAi knockdown of the *5HT2b* gene in dFB neurons to test the necessity of *5HT2b*. A dFB gene knockdown strain showed shortened sleep duration and impaired sleep homeostasis (*Figure 6K–O*). These results indicate that 5HT2b in dFB neurons is sufficient to regulate sleep homeostasis.

## Requirement for *5HT2b* gene in a pair of dFB neurons for sleep homeostasis

To determine whether a single pair of dFB neurons that were positive for the *5HT2b* receptor were necessary for sleep homeostasis, we used 5HT2b$^{GKO}$ to drive the expression of 5HT2b cDNA in all 5HT2b neurons, but with 23E10-LexA to drive the expression of LexAop-Gal80, thus suppressing *5HT2b* expression in only one pair of dFB neurons (*Figure 6F*). Dual reporters for 23E10 and 5HT2b were used to test the efficiency of Gal80 suppression, which revealed a single pair of neurons that were positive for both 23E10 and 5HT2b (*Figure 6—figure supplement 1D and E*), but no double-positive neurons in the presence of LexAop-Gal80 (*Figure 6—figure supplement 1F and G*), indicating that the Gal4 was efficiently suppressed in the dFB by LexA-Gal80. This allowed us to express *5HT2b* in all *5HT2b*-positive neurons except this single pair of dFB neurons. The sleep duration of flies in which 5HT2b$^{GKO}$ drove the expression of *5HT2b* cDNA in all 5HT2b neurons. except one pair of dFB neurons, in the *5HT2b* knockout background was significantly decreased, to a level similar to that of the *5HT2b* knockout mutants (*Figure 6G and H*). Furthermore, flies lacking *5HT2b* receptor in dFB neurons had no sleep recovery after 12-hr deprivation (*Figure 6I and J*). In addition, we rescued the expression of *5HT2b* with Gal80 in dFB neurons in the background of *5HT2b* knockdown to test the sufficiency of *5HT2b*. Both sleep duration and homeostasis were restored in this strain (*Figure 6K–O*). These results indicate that *5HT2b* in a small subset, probably a single pair, of dFB neurons is necessary for the regulation of sleep homeostasis.

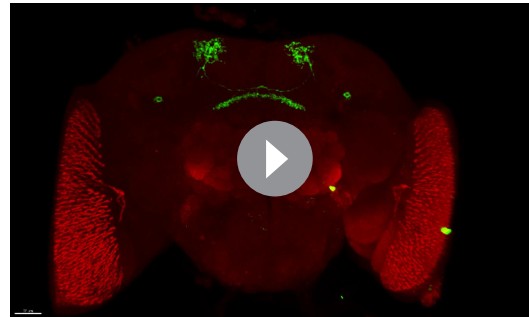

**Video 3.** 23E10 and 5HT2b intersectional neurons. Two pairs of 23E10 and 5HT2b intersectional neurons in the brain.

DOI: https://doi.org/10.7554/eLife.26519.017

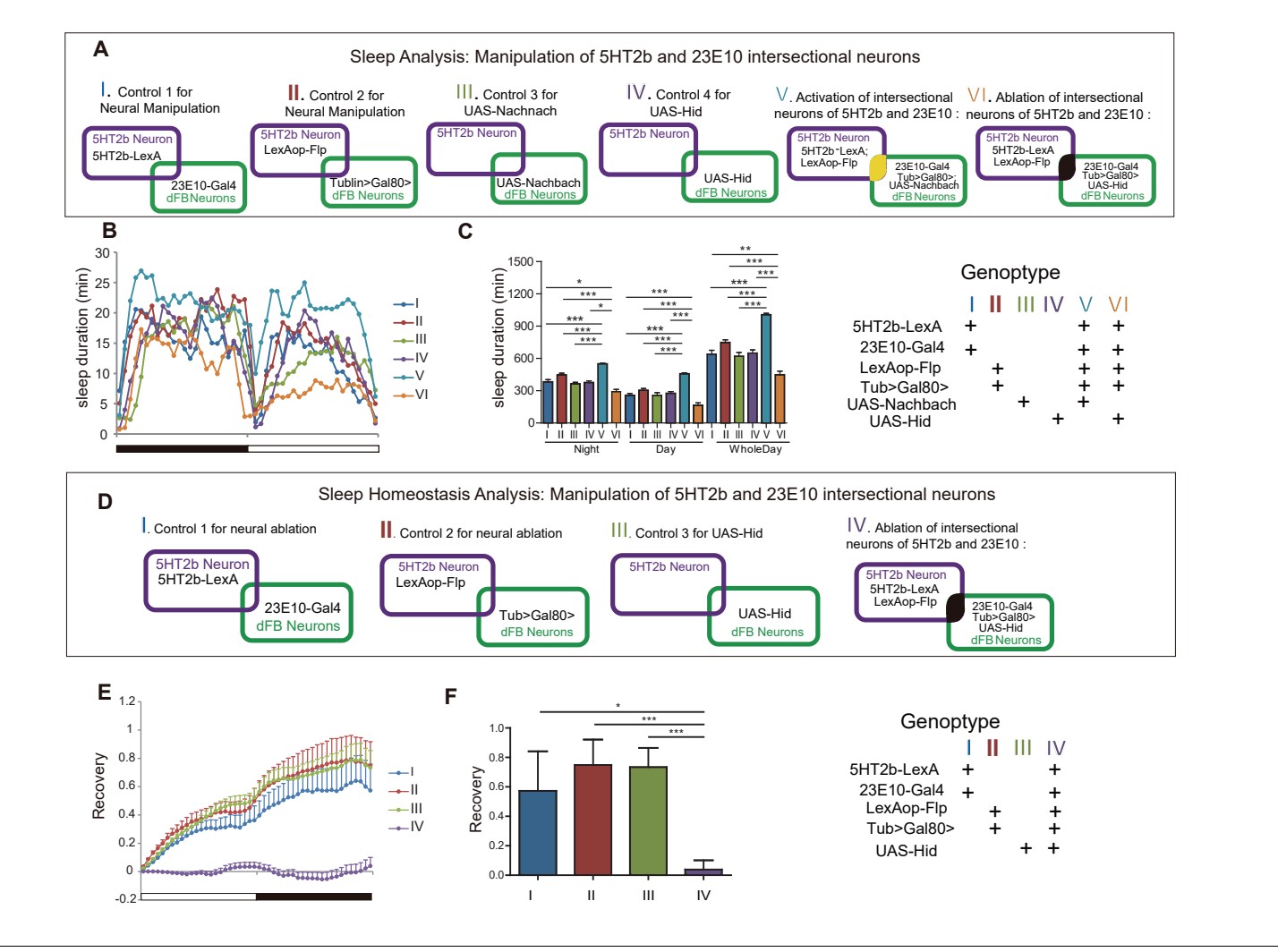

**Figure 5.** Effect of manipulating 5HT2b and dFB intersectional neurons on sleep. (A) Schematic illustration of the genetic manipulation of 5HT2b and dFB neurons for sleep analysis. (B–C) Activation of 5HT2b and 23E10 intersectional neurons increased sleep, and ablation of these neurons reduced sleep. Sleep profiles for flies with 23E10 neuron activation and control flies for 24 hr (B). Flies with neural activation slept more whereas those with ablation slept less . (C). Mean ± SEM, n = 48 for I, n = 48 for II, n = 48 for III, n = 48 for IV, n = 47 for V and n = 41 for VI flies. (D) Schematic illustration of genetic manipulation of 5HT2b and dFB neurons for sleep homeostasis analysis. (E–F) Ablation of 23E10 neurons impaired sleep rebound. Flies with 5HT2b and 23E10 intersectional neuron ablation had abnormal recovery rate after 12-hr sleep deprivation (mean ± SEM, n = 32 for I, n = 35 for II, n = 44 for III, and n = 40 for IV). Statistical analysis was performed with one-way ANOVA: *p<0.05, **p<0.01, ***p<0.001.

DOI: https://doi.org/10.7554/eLife.26519.018

The following figure supplement is available for figure 5:

**Figure supplement 1.** Effect of manipulating 5HT2b and dFB intersectional neurons on sleep.
DOI: https://doi.org/10.7554/eLife.26519.019

## Discussion

Our results indicate that the 5HT2b receptor is required in one pair of dFB neurons to regulate sleep homeostasis. Our finding of the differential roles of the receptors in promoting sleep and in regulating sleep recovery clearly show that the total amount of sleep and sleep recovery after deprivation may involve different mechanisms. dFB neurons have been shown to be important in sleep (*Donlea et al., 2014*). Our results not only reveal that serotonergic signaling is crucial but also show that the expression of *5HT2b* in a very small subset of dFB neurons is necessary and sufficient and explain the function of the 5HT2b receptor in sleep regulation. The intersectional strategy has shown 2–4 pairs of neurons that are positive for both 5HT2b and 23E10 but only one pair is shown with the

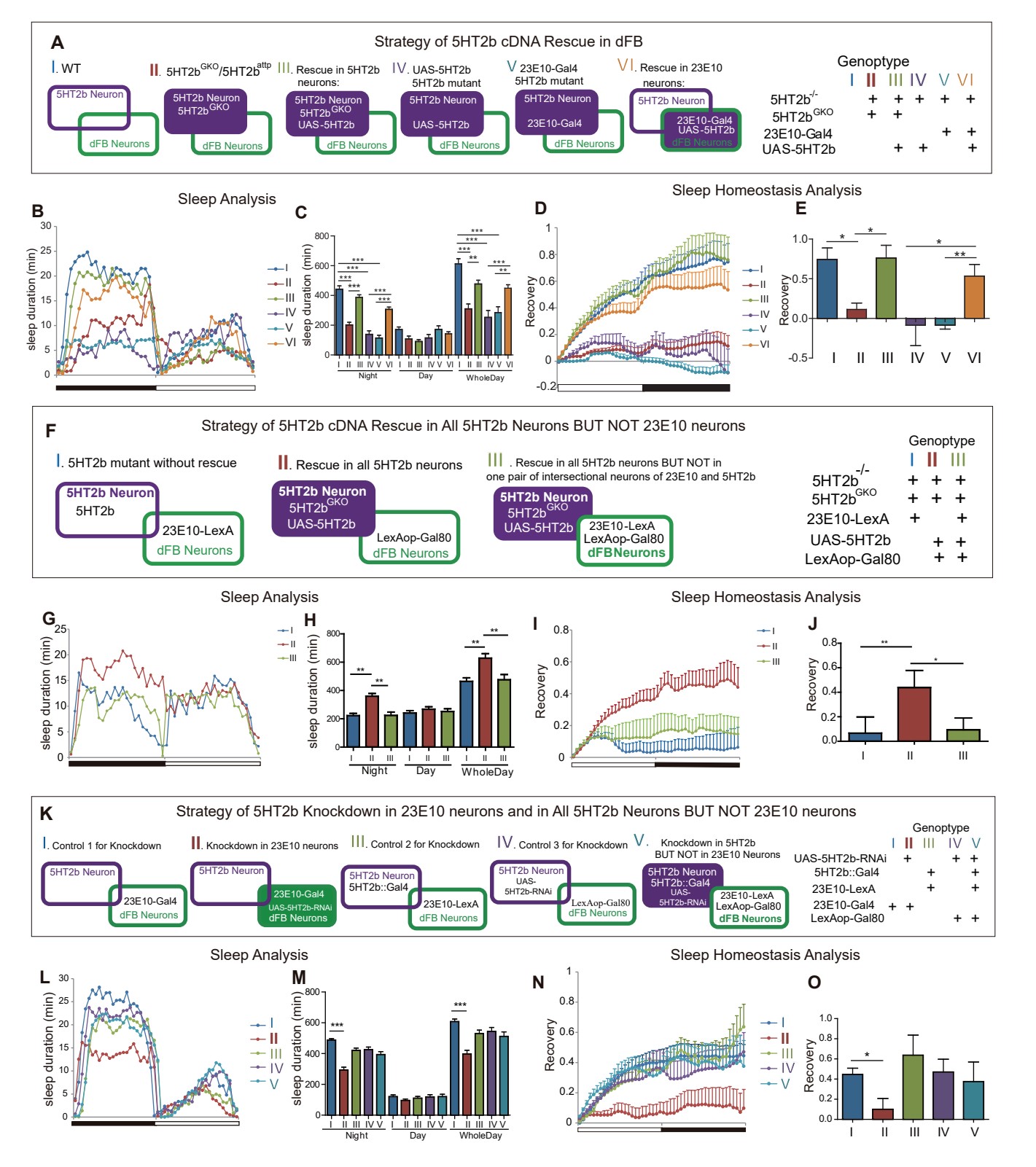

**Figure 6.** Regulation of sleep and sleep homeostasis by the *5HT2b* gene in one pair of dFB neurons. (A) Schematic illustration of genetic rescue with *5HT2b* cDNA in *5HT2b* mutants. (B) Sleep profile for 5HT2b homozygous heterzygous mutants and genetic rescue lines with 5HT2b^GKO or 23E10-Gal4. (C) *5HT2b* mutants slept less than wt or genetic rescue stains with 5HT2b^GKO or 23E10-Gal4 (mean ± SEM, n = 48 for I, n = 48 for II, n = 48 for III, n = 848 for IV, n = 32 for V and n = 48 for VI). (D and E) Heterozygous mutants and flies with *5HT2b* cDNA restored in 5HT2b or 23E10 neurons

*Figure 6 continued on next page*

*Figure 6 continued*

had normal recovery rate after 12-hr sleep deprivation (**D**). (**E**) Statistical analysis (mean ± SEM, n = 43 for I, n = 47 for II, n = 36 for III, n = 25 for IV, n = 31 for V and n = 33 for VI). (**F**) Schematic illustration of genetic rescue in 5HT2b but not 23E10 neurons. (**G**) Sleep profile for the *5HT2b* homozygous mutant and genetic rescue in 5HT2b neurons and genetic rescue lines in all 5HT2b neurons but not in 23E10 neurons. 5HT2b mutant flies and flies with 5HT2b rescue in all 5HT2b neurons but not in 23E10 neurons were similar in sleep duration during both days and nights, but slept less than genetic rescue strains with 5HT2bGKO in night-time sleep (**G**). (**H**) Statistical analysis (mean ± SEM, n = 48 for I, 48 for II, and 46 for III, respectively). (**I–J**) *5HT2b* mutant flies and flies with *5HT2b* rescue in all 5HT2b neurons but not 23E10 neurons showed impaired sleep homeostasis (**I**). (**J**) Statistical analysis (mean ± SEM, n = 41 for I, n = 40 for II, and n = 43 for III). (**K**) Schematic illustration of *5HT2b* gene knockdown in 23E10 neurons or in all *5HT2b* neurons but not 23E10 neurons. (**L**) Sleep profile for strains of *5HT2b* gene knockdown 23E10 neurons and Gal80 rescue in 23E10 neurons. Gene knockdown in dFB neurons showed shortened sleep duration (II), and the Gal80 rescued strain (V) showed normal sleep duration. (**M**) Statistical analysis (mean ± SEM, n = 46 for I, n = 42 for II, n = 43 for III, n = 48 for IV and n = 44 for V). (**N–O**) Gene knockdown in dFB neurons showed abnormal sleep homeostasis, and the Gal80 rescued strain (V) was able to restore the sleep homeostasis. (**O**) Statistical analysis (mean ± SEM, n = 43 for I, n = 30 for II, n = 37 for III, n = 48 for IV and n = 35 for V). One-way ANOVA was used to detect statistical difference between different genotypes. *p<0.05, **p<0.01, ***p<0.001.
DOI: https://doi.org/10.7554/eLife.26519.020

The following figure supplement is available for figure 6:

**Figure supplement 1.** Appoaches for neural manipulation and genetic rescue.
DOI: https://doi.org/10.7554/eLife.26519.021

non-intersectional strategy. This could be due to the possibility that the intersectional strategy may involve protein expression at an immature developmental stage, whereas the drivers in the non-intersectional strategy directly showed expression in mature neurons. Other brain regions and neurotransmitters have been implicated in sleep regulation. Sleep is also regulated by dopamine and its G-protein-coupled receptor *DopR* in PPL1 and PPM3 clusters (*Liu et al., 2012*; *Ueno et al., 2012*). It will be interesting to investigate whether and how dopaminergic and serotonergic systems interact to regulate sleep or sleep homeostasis in drosophila. Two separate excitatory synaptic microcircuits oppositely regulate sleep, and the activity of the sleep-promoting microcircuit in the MB is increased after sleep deprivation (*Aso et al., 2014*; *Sitaraman et al., 2015*). A subset of EB neurons encode sleep drive by modulating calcium level and synaptic strength (*Liu et al., 2016*). It will also be interesting to investigate how different regions in the brain work with each other to regulate sleep.

Mutants exhibited normal circadian periods, but we could not rule out other circadian rhythm defects, such as circadian entrainment. It has been reported that *5HT1a* mutants have slight reduced recovery sleep after 6 hr of sleep deprivation (*Yuan et al., 2006*). In this study, however, we found normal sleep rebound after 12-hr sleep deprivation. One possibility that might explain this discrepancy is that we used different *5HT1a* mutants. In the sleep-recovery experiment, we observed no significant difference between wt flies fed with mock or 5HTP, which indicated that wt 5-HT level was probably saturated for sleep recovery. Although the daytime sleep of mutant flies with 5HTP rescue was higher than that of the wt, there was still room for increase if the recovery ability of the flies is restored. Therefore sleep rebound could be dissociated from the baseline sleep. In mammals, the role of 5-HT in sleep is controversial. Early results from lesion and neuropharmacological studies had implicated 5-HT in the initiation and maintenance of slow wave sleep (SWS) (*Jouvet, 1972*), but other reports challenged the serotonergic hypothesis of sleep promotion (*Monti, 2011*). Recent papers claimed that the observed changes in sleep were indirect as resulted from 5-HT involvement in thermoregulation when all serotonergic neurons were genetically deleted in mice (*Buchanan and Richerson, 2010*; *Hodges et al., 2008*). Our unpublished data revealed that sleep was decreased in Tph2$^{-/-}$ mice, in which serotonin could not be synthesized in the brain but with normal thermoregulation (Yan HM and YR). However, it has not been studied whether 5-HT is important for sleep rebound after deprivation in mammals.

The ventrolateral preoptic nucleus (VLPO) is thought to be important for sleep control in mammals. Its activity correlates with the sleep/wake cycle (*Chou et al., 2002*; *Gaus et al., 2002*; *Ko et al., 2003*; *Lu et al., 2000*; *Sherin et al., 1998*; *Sherin et al., 1996*). Interestingly, both the mammalian VLPO and the drosophila dFB are sensitive to isoflurane, a general hypnotic anesthetic, with the dFB known to show higher sensitivity to isoflurane after sleep deprivation (*Kottler et al., 2013*; *Lu et al., 2008*; *Moore et al., 2012*; *Nelson et al., 2002*). Both the VLPO and the dFB increase their firing rates in response to sleep deprivation (*Alam et al., 2014*; *Donlea et al., 2014*). Furthermore, the VLPO contains a subset of sleep-promoting neurons that are excitable by 5-HT,

and another subset of neurons that are inhibitable by 5-HT (*Gallopin et al., 2005*). It is not known which 5-HT receptors in the VLPO mediated these responses. In mice, 5HT2A and 5HT2C receptors are homologs of the drosophila 5HT2b receptor. Like drosophila *5HT2b* mutants, 5HT2C receptor knock-out mice had more wakefulness and abnormalities in rapid eye movement (REM) sleep (*Frank et al., 2002*). *5HT2A* mutant mice exhibited increased wakefulness and reduced non-rapid eye movement (NREM) sleep (*Popa et al., 2005*). It will be interesting to examine the VLPO distributions of 5HT2C and 5HT2A receptors in mammals and to investigate their roles in sleep homeostasis.

Our results provided genetic and neural illustrations of how serotonin affects sleep regulation. Further studies are required to elucidate the neurochemistry and physiology of dFB neurons in sleep regulation and to refine the downstream effectors of dFB neurons. It will be of great interest to determine whether the serotonin in dFB that regulates sleep also influences memory formation.

## Materials and methods

### Drosophila stocks and culture

Flies were reared at 25°C and 60% humidity and were kept in a 12-hr light-12-hr dark cycle (or in constant darkness when specified). For thermogenetic activation of neurons, flies were raised and loaded into the visual-recording system at 22°C. Temperature was changed to 28°C when neural activation to place. Flies were backcrossed into a Canton S for at least five generations. UAS-Hid, Dilp2-Gal4, Pdf-Gal4 and MB247-Gal4 were generously provided by Aike Guo (Institute of Biophysics, CAS). Tub > Gal80 > was provided by Jing Wang (UCSD). UAS-StingerGFP and LexAop-tdTomato were gifts from Barry Dickson (Janelia Research Campus, HHMI). UAS-TrpA1 was provided by Paul Garrity (Brandeis University). UAS-mCD8GFP, UAS-StingerGFP, UAS-DscamGFP, UAS-sytGFP, UAS-DenMark, UAS-NachBach, UAS-FRT-stop-FRT-mCD8GFP, 23E10-Gal4, 23E10-LexA, 69F08-Gal4, 18H11-Gal4, LexAop-Gal80, TH-Gal4 and LexAop-Flp were obtained from the Bloomington Stock Center. UAS-5HT2b-RNAi (KK102356) was obtained from the Vienna Drosophila Resource Center (VDRC). *Table 1* contains the complete genotypes used in the figures.

### Pharmacological treatment of flies with 5-HTP

5-HTP was obtained from Sigma. Wt or *Trh* mutant flies were raised with normal food during the larval stage and immediately transferred to 5HTP food (5% sucrose, 2% agar and 2 mg/ml 5HTP) after eclosion. For immunostaining analysis, after three days of feeding with 5HTP-containing food, flies were dissected for immunohistochemistry. For behavioral analysis, flies were fed with 5HTP food after eclosion (usually 3–5 days), then transferred to glass tubes with 5HTP food for sleep analysis.

### Construction of transgenic, knockout and knockin lines of drosophila

The UAS-5HT2b DNA construct was generated by subcloning *5HT2b* cDNA into the pACU2 vector, with the *5HT2b* cDNA from the Vosshall lab at Rockefeller University and the pACU2 vector from the Jan Lab at UCSF (*Gasque et al., 2013*; *Han et al., 2011*). Superfoder GFP cDNA was subcloned from the Addgene plasmid (No.60904).

The end-out method was used to generate mutant lines (*Huang et al., 2009*). Gal4 was introduced to replace the gene by homologous recombination. In each of the targeting constructs, the entire first exon with the first ATG codon of the gene (or the entire gene) was deleted. At least the first transmembrane domains of the seven transmembrane domains in a receptor were also deleted to ensure null mutants. The 5' homologous arms (~5 KB) and the 3' homologous arm (~3 KB) were inserted into pGX2 vectors by homologous recombination repair. We named the receptor mutants using Strategy I as 5HT1a$^{Gal4}$, 5HT1b$^{Gal4}$, 5HT2a$^{Gal4}$, 5HT2b$^{Gal4}$ and 5HT7$^{Gal4}$. We also introduced loxp and attp sites along with Gal4, which enabled us to flip out the introduced sequence when crossing with a cre strain, leaving a functional attp site in the gene targeted region (*Figure 1B*). We named the *5HT2b* mutant with introduced sequence flipped out 5HT2b$^{attp}$ (*Figure 2G–H* and *Figure 6A–J*).

The Crispr/Cas9 system was used to generate *Trh* indel mutant flies. The gRNA and Cas9 mRNA were designed and generated as described previously (*Yu et al., 2013*). The gRNA targeted the

**Table 1.** Complete genotypes used in the figures.
This table contains the genotypes of flies used in imaging and behavioral analysis.

| *Figure 1* | |
| --- | --- |
| E–J | Iso-CS |
| | w+;5HT1a$^{Gal4}$;+ |
| | w+;5HT1b$^{Gal4}$;+ |
| | w+;+;5HT2a$^{Gal4}$ |
| | w+;+;5HT2b$^{Gal4}$ |
| | w+;+;5HT7$^{Gal4}$ |
| | w+;+;Trh$^{01}$ |
| | w+;+;Trh$^{GKO}$ |
| ***Figure 2*** | |
| A,B | Iso-CS |
| | w+;5HT1a$^{Gal4}$;+ |
| | w+;5HT1b$^{Gal4}$;+ |
| | w+;+;5HT2a$^{Gal4}$ |
| | w+;+;5HT2b$^{Gal4}$ |
| | w+;+;5HT7$^{Gal4}$ |
| | w+;+;Trh$^{01}$ |
| | w+;+;Trh$^{GKO}$ |
| C,D | Iso-CS(Mock) |
| | Iso-CS(5HTP) |
| | w+;+;Trh$^{01}$(Mock) |
| | w+;+;Trh$^{01}$(5HTP) |
| | w+;+;Trh$^{GKO}$(Mock) |
| | w+;+;Trh$^{GKO}$(5HTP) |
| E, H–J | w−;+;UAS-TrpA1/+ |
| | w-−;+;TH-Gal4/+ |
| | w−;+;UAS-TrpA1/TH-Gal4 |
| F, H–J | w−;+;UAS-TrpA1,Trh$^{01}$/Trh$^{01}$ |
| | w−;+;TH-Gal4,Trh$^{01}$/Trh$^{01}$ |
| | w−;+;UAS-TrpA1,Trh$^{01}$/TH-Gal4,Trh$^{01}$ |
| G, H–J | w−;+;UAS-TrpA1,5HT2b$^{attp}$/5HT2b$^{attp}$ |
| | w−;+;TH-Gal4,5HT2b$^{attp}$/5HT2b$^{attp}$ |
| | w−;+;UAS-TrpA1,5HT2b$^{attp}$/TH-Gal4,5HT2b$^{attp}$ |
| ***Figure 3*** | |
| A | w−;UAS-mCD8GFP/+;Trh::Gal4/+ |
| B | w−;UAS-stingerGFP/+;Trh::Gal4/+ |
| C | w−;UAS-DscamGFP/+;Trh::Gal4/+ |
| D | w−;UAS-sytGFP/+;Trh::Gal4/+ |
| E | w−;UAS-mCD8GFP/+;5HT2b::Gal4/+ |
| F | w−;UAS-stingerGFP/+;5HT2b::Gal4/+ |
| G | w−;UAS-DscamGFP/+;5HT2b::Gal4/+ |
| H | w−;UAS-sytGFP/+;5HT2b::Gal4/+ |
| ***Figure 4*** | |
| A | w−;+;UAS-5HT2b-RNAi/+ |

*Table 1 continued on next page*

*Table 1 continued*

*Figure 1*

| | |
|---|---|
| | w−;+;5HT2b::Gal4/+ |
| | w−;+;5HT2b::Gal4/UAS-5HT2b-RNAi |
| | w−;+;MB247-Gal4/+ |
| | w−;+;MB247-Gal4/UAS-5HT2b-RNAi |
| | w−;+;R69F08-Gal4/+ |
| | w−;+;R69F08-Gal4/UAS-5HT2b-RNAi |
| | w−;+;pdf-Gal4/+ |
| | w−;+;pdf-Gal4/UAS-5HT2b-RNAi |
| | w−;+;dilp2-Gal4/+ |
| | w−;+;dilp2-Gal4/UAS-5HT2b-RNAi |
| | w−;+;R18H11-Gal4/+ |
| | w−;+;R18H11-Gal4/UAS-5HT2b-RNAi |
| | w−;+;R23E10-Gal4/+ |
| | w−;+;R23E10-Gal4/UAS-5HT2b-RNAi |
| B | w−;UAS-mCD8GFP/+;23E10-Gal4/+ |
| C | w−;UAS-DenMark,UAS-sytGFP/+;23E10-Gal4/+ |
| D,E | w−;UAS-FRT-stop-FRT-mCD8GFP/+; 5HT2b-LexA,23E10-Gal4/LexAop-Flp |
| F–H | w−;UAS-stingerGFP,LexAop-TdTomato/+;5HT2b-LexA,23E10-Gal4/+ |
| I–K | w−;+;5HT2b$^{sfGFP}$/+ |
| L–N | w−;UAS-5HT2b$^{sfGFP}$/+;23E10-Gal4/+ |

*Figure 5*

| | |
|---|---|
| B–C(I) | w−; +;5HT2b-LexA,23E10-Gal4/+ |
| B–C (II) | w−;Tub > Gal80>/+;LexAop-Flp/+ |
| B–C (III) | w−;UAS-Nachbach/+;+ |
| B–C (IV) | w−;UAS-Hid/+;+ |
| B–C (V) | w−;Tub > Gal80>,UAS-Nachbach/+;5HT2b-LexA,23E10-Gal4/LexAop-Flp |
| B–C (VI) | w−;Tub > Gal80>,UAS-Hid/+;5HT2b-LexA,23E10-Gal4/LexAop-Flp |
| E–F(I) | w−;+;5HT2b-LexA,23E10-Gal4/+ |
| E–F(II) | w−;+;Tub > Gal80>/+;Lexop-Flp/+ |
| E–F(III) | w−;UAS-Hid/+;+ |
| E–F(IV) | w−;Tub > Gal80>,UAS-Hid/+;5HT2b-LexA,23E10-Gal4/LexAop-Flp |

*Figure 6*

| | |
|---|---|
| B–E(I) | w+;+;+ |
| B–E(II) | w−;+;5HT2bGKO/5HT2battp |
| B–E(III) | w−;UAS-5HT2b/+;5HT2b$^{GKO}$/5HT2b$^{attp}$ |
| B–E(IV) | w−;UAS-5HT2b/+;5HT2b$^{attp}$/5HT2b$^{attp}$ |
| B–E(V) | w−;+;23E10-Gal4,5HT2b$^{attp}$/5HT2b$^{attp}$ |
| B–E(VI) | w−;UAS-5HT2b/+;23E10-Gal4,5HT2b$^{attp}$/5HT2b$^{attp}$ |
| G–J(I) | w−;23E10-LexA/+;5HT2b$^{GKO}$/5HT2b$^{attp}$ |
| G–J(II) | w−;UAS-5HT2b/+;5HT2b$^{GKO}$,LexAop-Gal80/5HT2b$^{attp}$ |

*Table 1 continued on next page*

*Table 1 continued*

| *Figure 1* | |
|---|---|
| G–J(III) | w−;UAS-5HT2b/23E10-LexA;5HT2b$^{GKO}$,LexAop-Gal80/5HT2b$^{attp}$ |
| L–O(I) | w−;+;23E10-Gal4/+ |
| L–O(II) | w−;UAS-5HT2b-RNAi/+;23E10-Gal4/+ |
| L–O(III) | w−;23E10-LexA/+;5HT2b::Gal4/+ |
| L–O(IV) | w−;LexAop-Gal80/+;UAS-5HT2b-RNAi/+ |
| L–O(V) | w−;LexAop-Gal80/23E10-LexA;5HT2b::Gal4/UAS-5HT2b-RNAi |
| *Figure 1—figure supplement 2* | |
| A | w+;+;+ |
| B | w-;+;Trh::Gal4 |
| C | w+;+;Trh$^{01}$ |
| D | w+;+;Trh$^{GKO}$ |
| E | w+;+;Trh$^{01}$ |
| F | w+;+;Trh$^{GKO}$ |
| G | w+;+;Trh$^{01}$ |
| H | w+;+;Trh$^{GKO}$ |
| I-J | isoCS |
| | w+;+;Trh$^{01}$ |
| *Figure 1—figure supplement 3* | |
| A–F | Iso-CS |
| | w+;5HT1a$^{Gal4}$;+ |
| | w+;5HT1b$^{Gal4}$;+ |
| | w+;+;5HT2a$^{Gal4}$ |
| | w+;+;5HT2b$^{Gal4}$ |
| | w+;+;5HT7$^{Gal4}$ |
| | w+;+;Trh$^{01}$ |
| | w+;+;Trh$^{GKO}$ |
| *Figure 1—figure supplement 4* | |
| A–F | Iso-CS |
| | w+;5HT1a$^{Gal4}$;+ |
| | w+;5HT1b$^{Gal4}$;+ |
| | w+;+;5HT2a$^{Gal4}$ |
| | w+;+;5HT2b$^{Gal4}$ |
| | w+;+;5HT7$^{Gal4}$ |
| | w+;+;Trh$^{01}$ |
| | w+;+;Trh$^{GKO}$ |
| *Figure 2—figure supplement 1* | |
| A–B | Iso-CS |
| | w+;5HT1a$^{Gal4}$;+ |
| | w+;5HT1b$^{Gal4}$;+ |
| | w+;+;5HT2a$^{Gal4}$ |
| | w+;+;5HT2b$^{Gal4}$ |
| | w+;+;5HT7$^{Gal4}$ |
| | w+;+;Trh$^{GKO}$ |

*Table 1 continued on next page*

*Table 1 continued*

| | |
|---|---|
| *Figure 1* | |
| *Figure 2—figure supplement 2* | |
| A–B | Iso-CS |
| | w+;5HT1a$^{Gal4}$;+ |
| | w+;5HT1b$^{Gal4}$;+ |
| | w+;+;5HT2a$^{Gal4}$ |
| | w+;+;5HT2b$^{Gal4}$ |
| | w+;+;5HT7$^{Gal4}$ |
| | w+;+;Trh$^{01}$ |
| *Figure 2—figure supplement 3* | |
| B | Iso-CS |
| | w+;+;5HT2b$^{Gal4}$ |
| | w+;+;Trh$^{01}$ |
| C | Iso-CS |
| | w+;5HT1a$^{Gal4}$;+ |
| | w+;5HT1b$^{Gal4}$;+ |
| | w+;+;5HT2a$^{Gal4}$ |
| | w+;+;5HT2b$^{Gal4}$ |
| | w+;+;5HT7$^{Gal4}$ |
| | w+;+;Trh$^{01}$ |
| | w+;+;Trh$^{GKO}$ |
| *Figure 3—figure supplement 1* | |
| A | w+;UAS-mCD8GFP/+;Trh::Gal4/+ |
| B | w+;UAS-mCD8GFP/+;Trh$^{GKO}$/+ |
| C | w+;UAS-mCD8GFP/+;5HT2b::Gal4/+ |
| D | w+;UAS-mCD8GFP/+;5HT2b$^{GKO}$/+ |
| E | w+;UAS-mCD8GFP/+;5HT2b-LexA/+ |
| F | w−;UAS-mCD8GFP/+;Trh::Gal4/+ |
| G | w−;UAS-stingerGFP/+;Trh::Gal4/+ |
| H | w−;UAS-DscamGFP/+;Trh::Gal4/+ |
| I | w−;UAS-sytGFP/+;Trh::Gal4/+ |
| J | w−;UAS-mCD8GFP/+;5HT2b::Gal4/+ |
| K | w−;UAS-stingerGFP/+;5HT2b::Gal4/+ |
| L | w−;UAS-DscamGFP/+;5HT2b::Gal4/+ |
| M | w−;UAS-sytGFP/+;5HT2b::Gal4/+ |
| *Figure 4—figure supplement 1* | |
| A | w−;+;5HT2b::Gal4/+ |
| | w−;+;UAS-5HT2b-RNAi |
| | w−;+;5HT2b::Gal4/UAS-5HT2b-RNAi |
| B | w−;UAS > Stop > mCD8GFP/+;Lexop-Flp,5HT2b-LexA/MB247-Gal4 |
| | w−;UAS > Stop > mCD8GFP/+;Lexop-Flp,5HT2b-LexA/R69F08-Gal4 |
| | w−;UAS > Stop > mCD8GFP/+;Lexop-Flp,5HT2b-LexA/Pdf-Gal4 |
| | w−;UAS > Stop > mCD8GFP/+;Lexop-Flp,5HT2b-LexA/Dilp2-Gal4 |

*Table 1 continued on next page*

*Table 1 continued*

*Figure 1*

| | |
|---|---|
| | w−;UAS > Stop > mCD8GFP/+;Lexop-Flp,5HT2b-LexA/R18H11-Gal4 |
| | w−;UAS > Stop > mCD8GFP/+;Lexop-Flp,5HT2b-LexA/R23E10-Gal4 |

*Figure 5—figure supplement 1*

| | |
|---|---|
| A–B | w−;23E10-Gal4/+;+ |
| | w−;UAS-Nachbach/+;+ |
| | w−;23E10-Gal4/UAS-Nachbach;+ |
| C | w−;23E10-Gal4/UAS-stingerGFP;+ |
| D | w−;23E10-Gal4/UAS-Hid,UAS-stingerGFP;+ |
| E–H | w−;23E10-Gal4/+;+ |
| | w−;UAS-Hid/+;+ |
| | w−;23E10-Gal4/UAS-Hid;+ |

*Figure 6—figure supplement 1*

| | |
|---|---|
| B | w−;Tub > Gal80>,UAS-mCD8GFP/+;5HT2b-LexA,23E10-Gal4/LexAop-Flp |
| C | w−;Tub > Gal80>,UAS-mCD8GFP/+;23E10-Gal4/+ |
| E | w−;UAS-stingerGFP,LexAop-tdTom/23E10-LexA;5HT2b$^{GKO}$/+ |
| F | w−;UAS-stingerGFP,LexAop-tdTom/23E10-LexA; |

5HT2b$^{GKO}$,LexAop-Gal80/+

DOI: https://doi.org/10.7554/eLife.26519.022

catalytic center of the Trh enzyme. We named the *Trh* indel mutant Trh$^{01}$. PCR analysis and DNA sequencing were used for further confirmation.

Two different gRNAs were used to generate deletion in genome regions of interest. The sgRNA sequence was subcloned into a U6b-sgRNA vector to generate gRNA in fly embryos (*Ren et al., 2013*). The nos-Cas9 flies were used for injection. The 5' homologous arms (~2.5 KB) and the 3' homologous arm (~3 KB) were inserted into a pBSK vector for homologous recombination repair. Arm fragments were amplified from the wt fly genome by Phanta Max DNA polymerase (Vazyme Biotech Co. Ltd.). The Gal4, Flp or LexA effector sequence was introduced between the two homologous arms and 3p3-RFP was used as the fluorescent marker for screening (*Xue et al., 2014a, 2014b*). We named the *Trh* with Gal4 knockin using Strategy II strain Trh$^{GKO}$, and the *Trh* Gal4 knockin using Strategy III strain Trh::Gal4. We named the 5HT2b with Gal4 knockin using Strategy II strain 5HT2b$^{GKO}$, the 5HT2b LexA knockin using Strategy II strain 5HT2b-LexA, and the Gal4 knockin using Strategy III strain 5HT2b::Gal4. PCR analysis and DNA sequencing were used for further confirmation. *Table 2* contains the primers and sgRNAs used.

## qRT-PCR

For each sample, 20 flies were ground on ice. RNA and cDNA were generated with the PrimerScript RT Master Mix (Takara). qRT–PCR was performed using TransStart Green qPCR SuperMix (Transgen Biotech) and the 96-well Applied Biosystems 7900HT Real-Time PCR System.

## Sleep analysis

Female or male flies aged 4–6 days were transferred into monitor tubes (5 mm × 65 mm) containing normal food. 48 flies were usually used in each group in our experiments. Female flies were used for sleep analysis if not specifically mentioned. Light was on from 9:00am to 9:00pm. Locomotor activity was monitored in the LD cycle using a video system (*Gilestro and Cirelli, 2009*; *Yi et al., 2013*; *Zimmerman et al., 2008*). Infrared LEDs were used for constant illumination for video recording. Images were acquired every second (sec) and processed by Visual C++ to determine the location of

**Table 2.** Primers and sgRNA.
The table contains the primers and sgRNA used in line construction.

| | | | |
|---|---|---|---|
| 5HT1a | 3' arm | For | ATGCTAGCGGCCGCGAGATTCGGA GGGAAAAGAATGTG |
| | | Rev | TACCATGGTACCTTTGTGGATACTCG GTGTGTTTTTTTG |
| | 5' arm | For | AAGTCGACTAGTAGCCAAAGCCCAAACATAGA |
| | | Rev | TTCCATGGCGCGCCCCAACGAGCCATTTGAGATT |
| 5HT1b | 3' arm | For | ATGCTAGCGGCCGCTGGCATAGTTTTTCGCTGTG |
| | | Rev | TACCATGGTACCCTCGCTTGTATATATGTA TATGTATATC |
| | 5' arm | For | AAGTCGACTAGTCCATATCGCCGACAAAAACT |
| | | Rev | TTCCATGGCGCGCCGTTGGCATTTGTTTGGCTTT |
| 5HT2a | 3' arm | For | ATGCTAGCGGCCGCCATGGTCAACATGGGTTCAA |
| | | Rev | TACCATGGTACCTTCATTCTAAAGCTGTGGGGC |
| | 5' arm | For | TTCCATGGCGCGCCAAACTTGTTGGCCTTGTTGG |
| | | Rev | TACTGCAGGCCTGCGCAATTGCTTAACAGACA |
| 5HT2b | 3' arm | For | ATGCTAGCGGCCGCGAAATGTTGCCGTGAGTCT |
| | | Rev | TAGGATCCGCGGTCCGCGTTATTATTCCTATTGTG |
| | 5' arm | For | AAGTCGACTAGTTTCAAAGTCCGGAACAGAGG |
| | | Rev | TTCCATGGCGCGCCAAACTCCATTTTCCGCACAG |
| 5HT7 | 3' arm | For | ATGCTAGCGGCCGCAAGGACAAAGGCGACACATC |
| | | Rev | TACCATGGTACCCAGCAGTCTTTCAATGGTGG |
| | 5' arm | For | AAGTCGACTAGTCCCTAGGTTTCCGGAAGAAG |
| | | Rev | TTCCATGGCGCGCCACACATATCCCCGTCCACAT |
| Primers and sgRNAs used in strategies II and III | | | |
| Trh^GKO | | For | cggGGTACCATGAACAGTCTTTAGCCACG |
| | 5' arm | Rev | cgcggatccGCGGCCGC TCACCCTGCGTAACCAGGTG |
| | 3' arm | For | cgcggatccGGCGCGCC CACAACAGCCTGAATGTGAG |
| | | Rev | tccccgcggTGCTACATTAAATCCTCCCA |
| | gRNA | | GAATGTGCTCATGTACGGCTCGG |
| | | | GAGAGCATCCAACGGCCATTTGG |
| Trh::Gal4 | 5' arm | For | cggggtaccAAGTCTCGGCAGAAACGCCTC |
| | | Rev | taacaccggtgcggccgcATCTCCCTCCGCCGTAGAGTT |
| | 3' arm | For | taacaccggtggcgcgccTTCTCGGGACCACAGAGCAGT |
| | | Rev | tccccgcggAATTTCGGCAGGACACCTCTG |
| | gRNA | | GGACAGCAATGAAACCTTAAGGG |
| | | | ACAGAGCAGTATAATTCACTTGG |
| 5HT2b^GKO/LexA | 5' arm | | cggGGTACCATTTGTGCCGTGTCCTGTAT |
| | | | cgcggatccGCGGCCGCCTGCAGCCGGCGCTCCCAGG |
| | 3' arm | | cgcggatccGGCGCGCCTGCGGCTAATCTTAGTTGGA |
| | | | tccccgcggGCATGGGTCTTGTTTCGTTT |
| | gRNA | | GGGCATCCTTACGCTGGTGAAAGG |
| | | | GGATAGAAAAATGGAATCGCTGG |
| 5HT2b::Gal4 | 5' arm | For | cggggtaccCGAGCTGACCACAAAGCGTGC |
| | | Rev | taacaccggtgcggccgcTCTGCTCGGTCGCCAGGCACT |
| | 3' arm | For | taacaccggtggcgcgccAGGATTCCACTGCTCCGGTGC |

*Table 2 continued on next page*

| | | Rev | tccccgcggAGTTTCGCTCTGTGGAAACCG |
|---|---|---|---|
| | gRNA | | TAACAGACGCCCGTTAAACCGGG |
| | | | ATCCACACTGGCGCACTTGT GGG |
| Primers in qRT-PCR | | | |
| 5HT2b For | | | CCAACTACTTCCTTATGTCG |
| | | Rev | GATAAATATCTGTCCACGGA |
| Act42a | | For | CTCCTACATATTTCCATAAAAGATCCAA |
| | | Rev | GCCGACAATAGAAGGAAAAACTG |
| gRNA for Trh[01] | | | GGAGATTGGCCTTGCATCTTTGGG |

DOI: https://doi.org/10.7554/eLife.26519.023

the flies. Sleep was defined as no detectable movement for 5 min or longer. Sleep parameters were calculated using Matlab (MathWorks, Natick, MA). Sleep duration, sleep bout duration, sleep latency and wake activity were analyzed for each 12-hr period of LD and DD for each condition. Female flies were sleep deprived by the mechanical method using a steering engine, which randomly shook for 20 s in 3 min (*Shaw et al., 2002*). Sleep deprivation efficiency was recorded by a DAM system. Sleep loss was defined as total sleep duration last night. Individual sleep rebound was quantified hourly for 24 hr by dividing the cumulative amount of sleep regained by the sleep duration last night. Each experiment was repeated twice or three times.

## Circadian analysis

Male flies aged 3–6 days were housed individually in 65 mm glass tubes containing normal food. Locomotor activity was measured in TriKinetics DAM systems for 12 days in constant darkness. Periodogram analysis was completed with the Actogram J plugin (*Schmid et al., 2011*). Period length was determined by periodogram analysis.

## Confocal microscopy

Adult female flies of 5–10 days of age were collected and their CNS dissected in phosphate buffered saline (PBS). The brains were subject to 4% paraformaldehyde (wt/vol) fixation in the PBS for 2–4 hr and washed three times with the wash buffer (3% NaCl in 0.1% PBT) for 10 min at room temperature. Samples were transferred to 10% normal goat serum (vol/vol, diluted in 1% PBT) for overnight blocking at room temperature and incubated with antibodies (diluted in 10% normal goat serum and 0.1% PBT) at 4°C overnight. After washing samples three times for 15 min with wash buffer at room temperature and incubating samples with secondary antibody (1:500) at 4°C overnight, we mounted the samples with FocusClear (CelExplorer Labs CO) and imaged them on a Zeiss LSM710 confocal microscope. Images were processed using Imaris (Bitplane AG, Zurich, Switzerland, 640 RRID: SCR_007370). The following antibodies were used, chicken anti-GFP (1:1000; Abcam Cat# 13970, RRID:AB_300798), rabbit anti-RFP (1:2000; Clontech Cat# 632496, RRID: AB_10013483), rabbit anti-5-HT (1:2000; Immunostar Cat# 20080, RRID:AB_572263), mouse anti-nc82 (1:50; DSHB Cat# 2314866, RRID: AB_2314866), rat-anti Myc (1:500, Transgen). Secondary antibodies were AlexaFluor488 anti-rabbit (Life technologies Cat# A11008, RRID:AB_10563748), AlexaFluor488 anti-chicken (Life technologies Cat# A11039, RRID:AB_2534096), AlexaFluor546 anti-rabbit (Life technologies Cat# A11035, RRID:AB_2534093), and AlexaFluor633 anti-mouse (Life technologies Cat# A21052, RRID: AB_141459).

## Statistics

Pairwise comparisons were evaluated by Student's t test. ANOVA with Holm-Sidak corrections for multiple comparisons was used to test hypotheses involving multiple groups. Statistical analyses were performed with Prism 5 (GraphPad).

## Acknowledgements

We are grateful to Y Li for discussion of and comments on the manuscript, to J Ni, G Gao, B Zhang and R Jiao for help with Crispr/Cas9 technology, to S Waddel for sharing flies, to A Guo and W Yi for advice on sleep assays, to Y N Jan for pACU2 constructs, to L Vosshall for *5HT2b* cDNA, and to Peking-Tsinghua Center for Life Sciences, Beijing Advanced Innovation Center for Genomic Diagnosis, the Center for Instruments of School of Life Sciences in PKU for imaging facilitation, the NSFC 31000547 (JH), the National Natural Science Foundation of China (Project 31421003) and Beijing Municipal Natural Science Foundation (No. Z111107067311058 and Z151100003915121) for grant support. We thank W Yang and E Zhou for their support in design and softwares of video-based sleep analysis. We thank members in Y Rao's lab for helpful discussions.

## Additional information

### Funding

| Funder | Grant reference number | Author |
| --- | --- | --- |
| National Natural Science Foundation of China | Project 31421003 | Yi Rao |
| National Natural Science Foundation of China | Project 31000547 | Juan Huang |
| Beijing Municipal Natural Science Foundation | Z111107067311058 | Yi Rao |
| Beijing Municipal Natural Science Foundation | Z151100003915121 | Yi Rao |

The funders had no role in study design, data collection and interpretation, or the decision to submit the work for publication.

### Author contributions

Yongjun Qian, Conceptualization, Resources, Data curation, Software, Formal analysis, Validation, Investigation, Visualization, Methodology, Writing—original draft, Writing—review and editing; Yue Cao, Bowen Deng, Rui Xu, Dandan zhang, Resources, Data curation, Methodology; Guang Yang, Data curation, Methodology; Jiayun Li, Data curation, Validation, Investigation; Juan Huang, Resources, Data curation, Supervision, Funding acquisition; Yi Rao, Conceptualization, Formal analysis, Supervision, Funding acquisition, Validation, Investigation, Visualization, Methodology, Writing—original draft, Project administration, Writing—review and editing

### Author ORCIDs

Yongjun Qian [ORCID] http://orcid.org/0000-0002-2696-6730
Yi Rao [ORCID] http://orcid.org/0000-0002-0405-5426

### Decision letter and Author response

Decision letter https://doi.org/10.7554/eLife.26519.025
Author response https://doi.org/10.7554/eLife.26519.026

## Additional files

### Supplementary files

• Transparent reporting form
DOI: https://doi.org/10.7554/eLife.26519.024

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
