## [Decision Letter]

Thank you for submitting your article "Sleep Homeostasis Regulated by 5HT2b Receptor in a Single Pair of Neurons in the Dorsal Fan-Shaped Body of *Drosophila*" for consideration by *eLife*. Your article has been reviewed by three peer reviewers, and the evaluation has been overseen by a Reviewing Editor and a Senior Editor. The following individuals involved in review of your submission have agreed to reveal their identity: Kazuhiko Kume (Reviewer #2).

The reviewers have discussed the reviews with one another and the Reviewing Editor has drafted this decision to help you prepare a revised submission. Given the nature of the essential revisions described below, it may be best for you to respond with a plan and a timetable for the completion of this work. We will share this with the board member and reviewers and get back to you with recommendations.

Overall, the reviewers found your manuscript to provide an important and comprehensive analysis of the role of serotonin and different serotonin receptors in *Drosophila* sleep. In particular, localization of 5-HT2B effects on sleep to a pair of neurons in the fan-shaped body (FSB) would constitute a significant advance. However, the reviewers felt that the manuscript fell short of mapping 5HT2B function to the two FSB neurons. Necessity and sufficiency of 5HT2B in these neurons should be addressed through RNAi and rescue analysis respectively. The reviewers also recommend more comprehensive analysis of sleep rebound. In particular, they suggest inclusion of other methods of deprivation, such as activation of arousal promoting monoaminergic neurons, and also a weaker mechanical deprivation protocol (to exclude effects of serotonin on mechanical sensitivity). For all sleep homeostasis experiments, rebound measurements should be based on the actual amount of sleep lost by each individual fly, and post-rebound data should be evaluated to ensure a return to normal sleep levels.

For the broad readership of *eLife*, we also recommend that you include a schematic to depict the different brain regions assayed.

A number of clarifications and corrections are required:

Was 5HT2BGal4>5HT2B RNAi validated by RT-PCR or by testing whether RNAi reduces 5HT2BsfGFP signal?

Major changes in baseline are apparent in the paper and are somewhat of a concern, with 600 min in Figure 1, Figure 2, but much lower elsewhere (e.g. 400 min in Figure 5—figure supplement 1/1F). In addition, between Figure 1EF and Figure 1—figure supplement 2J, the daytime sleep phenotype of Trh01 appears different. Between Figure 2, the rebound pattern appears very different in the same wild type flies. In 2C, the rebound mainly occurred during the daytime while in 2E, rebound occurred at night. What accounts for this difference?

What was the time course of 5HTP treatment for experiments shown in Figure 2? The legend describes "Three day feeding of 2 mg/ml 5HTp" – did administration begin at the start of the baseline day and extend through the deprivation and recovery days? Or did it include 3 days prior to the baseline day? If the effects of 5HT on baseline sleep (Yuan et al., Curr Biol 16: 1051; 2006) were ramping up during the baseline day used to calculate sleep rebound, it would be impossible to dissociate the effects of 5HTP on baseline sleep and sleep rebound.

Also, do Figure 2/2F analyze the same populations of animals? The data look to be the same for most genotypes, but data for Trh01+5HTP appear to be different (<30% sleep recovery in Figure 2 vs. >40% sleep recovery in Figure 2). If Figure 2 is from a single experiment, this should be stated along with population sizes (n).

Subsection “5-HT Regulation of Sleep Recovery After Deprivation Through its 2b Receptor”: Figure 2—figure supplement 1 should include the 24 hr time profile data after sleep deprivation since the quantification was performed using 24 hr data in Figure 2 and others.

Necessary genetic controls are missing for Figure 5 (Undriven UAS-Nachbach and UAS-Hid), 5E-F (Undriven LexAop-Flp, Tub>Gal80> & UAS-Hid) and 6B-E (5HT2b mutants with gal4 drivers but no UAS rescue).

Figure 3 and Figure 3—figure supplement 1 appear to be nearly duplicate images (perhaps adjacent Z-sections). The authors should state this clearly so that readers do not get the impression that data are duplicated.

Subsection “Manipulation of 5HT2b and dFB intersectional neurons regulates sleep duration and homeostasis”, last paragraph: What was the efficiency of FLP-FRT recombination in the intersectional experiments in Figure 5?

Subsection “Manipulation of 5HT2b and dFB intersectional neurons regulates sleep duration and homeostasis”, last paragraph: The description of either Figure 5 or 5E is wrong. The control genotype III shows impaired sleep rebound.

Rescue manipulations in Figure 6 should probably have a wild-type (no transgene control), as the effects of overexpressing 5HT2B might not be apparent. Finally, some genotypes in Figure 6 are not fully clear. For animals positive for both "5HT2B mutant" and "5HT2B-GKO (genotype III), are these heterozygotes carrying two modified alleles at the 5HT2B locus?

Earlier work is not correctly cited in various parts of the manuscript (e.g. Becnel et al. 2011, Dierick et al. 2007 are cited as showing 5HT1a mutant flies have altered sleep, but these papers do not report these findings). In addition, the authors should discuss prior work pertinent to their study. For instance, the section on 5HTP feeding should refer to similar experiments done by Yuan et al. (2006) and discuss the possibility that increased sleep in the mutant background results from gain-of-function effects. In addition, findings with 5HT receptor and Trh GAL4 lines should be compared with those if earlier studies (Gnerer et al., 2015; Alekseyenko et al., 2010).

The design and structure of alleles is not adequately described. Primer sequences, deletion junctions, et cetera should be described to enable readers to assess the nature of these alleles and validate them independently.

[Editors' note: further revisions were requested prior to acceptance, as described below.]

Thank you for resubmitting your work entitled "Sleep Homeostasis Regulated by 5HT2b Receptor in a Single Pair of Neurons in the Dorsal Fan-Shaped Body of *Drosophila* " for further consideration at *eLife*. Your revised article has been re-evaluated by a Senior Editor, a Reviewing Editor, and three reviewers.

The manuscript has been improved but there are some remaining issues that need to be addressed before acceptance, as outlined below:

1) Genetic controls that were requested in the previous decision letter were not included:

"Necessary genetic controls are missing for Figure 5 (Undriven UAS-Nachbach and UAS-Hid), 5E-F (Undriven LexAop-Flp, Tub>Gal80> & UAS-Hid) & 6B-E (5HT2b mutants with gal4 drivers but no UAS rescue)."

Undriven UAS-Nachbac/+ and UAS-Hid/+ controls are required for proper interpretation of 5B-C. Data for these lines are included in Figure 5—figure supplement 1 but baseline sleep levels in the other controls between these panels do not appear to be consistent. If the undriven UAS controls from Figure 5—figure supplement 1 are directly compared to data in Figure 5, it is not clear that genotype IV (the expression of Hid in dFB neurons expressing 5HT2b) has a significant decrease in daytime or whole day sleep from UAS-Hid controls or that the night-time sleep in genotype III (expression of NachBac in dFB neurons expressing 5HT2b) is higher than the UAS-Nachbac controls in Figure 5—figure supplement 1. These results require proper genetic controls to support the author's interpretations.

Additionally, 23E10-GAL4 in a 5HT2b mutant background (with no UAS-5HT2b) is needed for Figure 6.

Rescue manipulations in Figure 6 should probably have a wild-type (no transgene control), as the effects of overexpressing 5HT2B might not be apparent. Finally, some genotypes in Figure 6 are not fully clear. For animals positive for both "5HT2B mutant" and "5HT2B-GKO (genotype III)", are these heterozygotes carrying two modified alleles at the 5HT2B locus?

No transgene control or overexpression genotypes have been included.

2) The interpretation of Figure 4 is not clear. These panels show that driving expression of 5HT2b::sfGFP in the dFB labels dFB neurons with GFP. If the authors want to show subcellular localization of 5HT2b in dFB neurons, the best strategy would be imaging 5HT2b::sfGFP protein-trap signal (see Figure 4) in flies that also contain R23E10-GAL4;UAS-RFP. That combination would permit clear colocalization with endogenous 5HT2b and dFB axons vs dendrites. Otherwise the authors should refrain from commenting on the subcellular location.

3) The manuscript and title state that 5HT2b functions in a "single pair of neurons", though the intersectional genetic strategies applied to 5HT2b access perhaps two pairs of neurons. The "single pair of neurons" is from a different Gal4/LexA overlap experiment (Figure 4), which was not used for experiments involving 5HT2b. This is not a sticking point for the conclusion that 5HT2B functions in a subset of 23E10 neurons, but there is no need to overstate the result in the title and paper.

4) There continue to be a number of errors throughout the revised manuscript (text and figures), including several mentioned in the first round of review. These include mismatched citations to figures in the text (Results), and typographical mistakes. For instance, in the subsection “Genotypes in figures” under Figure 4—figure supplement 1, FRT-flanked stop alleles are misspelled as "Strop". There are also inconsistencies in the nomenclature.

Note also:

Figure 1 are still not legible.

Figure 1 are out of order.

Figure 1—figure supplement 4: in all three panels, legend on x-axis is missing for several genotypes.

For Figure 2, the legend can be clarified to indicate that sleep gain is over 24h.

---

## [Author Response]

Overall, the reviewers found your manuscript to provide an important and comprehensive analysis of the role of serotonin and different serotonin receptors in Drosophila sleep. In particular, localization of 5-HT2B effects on sleep to a pair of neurons in the fan-shaped body (FSB) would constitute a significant advance. However, the reviewers felt that the manuscript fell short of mapping 5HT2B function to the two FSB neurons. Necessity and sufficiency of 5HT2B in these neurons should be addressed through RNAi and rescue analysis respectively. The reviewers also recommend more comprehensive analysis of sleep rebound. In particular, they suggest inclusion of other methods of deprivation, such as activation of arousal promoting monoaminergic neurons, and also a weaker mechanical deprivation protocol (to exclude effects of serotonin on mechanical sensitivity). For all sleep homeostasis experiments, rebound measurements should be based on the actual amount of sleep lost by each individual fly, and post-rebound data should be evaluated to ensure a return to normal sleep levels.

Necessity and sufficiency of 5HT2B in one pair of dFB neurons:

We have now used RNAi knockdown of 5HT2b gene in dFB neurons (UAS-5HT2b-RNAi driven by 23E10-Gal4,) to test the necessity of 5HT2b. In addition, we rescued the expression of 5HT2b gene with Gal80 (5HT2b::Gal4, 23E10-LexA, LexAop-Gal80, UAS-5HT2b-RNAi) in dFB neurons in the background of 5HT2b knockdown to test the sufficiency of 5HT2b. New results indicated that dFB gene knockdown strain (23E10-Gal4, UAS-5HT2b-RNAi) showed shortened sleep duration and impaired sleep homeostasis, and that 5HT2b expression in the dFB (5HT2b::Gal4, 23E10-LexA, LexAop-Gal80, UAS-5HT2b-RNAi) was able to rescue the sleep phenotype. We have therefore established the necessity and sufficiency of 5HT2b in the dFB to regulate sleep. The data were presented in Figure 6.

Analysis of sleep rebound:

We have now used additional methods of deprivation to study sleep homeostasis. We thermogenetically induced sleep loss by expressing transgenic TrpA1 channels (UAS-TrpA1) in dopaminergic neurons (TH-Gal4) which led to sleep deprivation, and we measured the sleep loss, sleep gain and recovery rate in wt, Trh mutant and 5HT2b mutant flies. New results indicated that Trh and 5HT2b mutant flies, after sleep deprivation induced by neural activation, showed abnormal sleep recovery comparing to wt flies (Figure 2). This method of deprivation, as suggested by the reviewer, is independent of mechanical stimulation, and thus verify the effect of Trh and 5HT2b in sleep rebound after deprivation.

Furthermore, we used weaker mechanical sleep deprivation protocol. Six-hour deprivation mechanical deprivation protocol (ZT18-ZT24) was used to supplement our previous method. Trh and 5HT2b mutants again showed impaired sleep recovery after 6-hour sleep deprivation (Figure 2—figure supplement 3).

Because it is difficult to trace the fly location during mechanical shakes with the video recording method, we used the DAM system to monitor sleep deprivation. We found that, in wt and mutant flies, sleep deprivation efficiencies were 100% (100% to wt, 100% to 5HT1a-/-, 100% to 5HT1b-/-, 99.3% to 5HT2a-/-, 99.9% to 5HT2b-/-, 100% to 5HT7-/- and 100% to Trh-/-) in Figure 2—figure supplement 2. It is reasonable to take the sleep amount of the immediately preceding unperturbed night as sleep loss during the deprivation. The revised manuscript showed the sleep loss and sleep gain in the thermogenetically induced sleep deprivation method.

For wt and mutant flies, we measured sleep recovery for 48 hours immediately after sleep deprivation. We found that sleep recovery took place in the first 24-hour after sleep deprivation, and no significant recovery in the second 24-hour (Figure 2—figure supplement 3).

For the broad readership of eLife, we also recommend that you include a schematic to depict the different brain regions assayed.The revised manuscript included a cartoon in Figure 4 to depict the different brain regions assayed.A number of clarifications and corrections are required:Was 5HT2BGal4>5HT2B RNAi validated by RT-PCR or by testing whether RNAi reduces 5HT2BsfGFP signal?

RNAi efficiency was validated by qRT-PCR (5HT2b::Gal4 > UAS-5HT2b-RNAi) (Figure 4—figure supplement 1).

Major changes in baseline are apparent in the paper and are somewhat of a concern, with 600 min in Figure 1, Figure 2, but much lower elsewhere (e.g. 400 min in Figure 5—figure supplement 1/1F).

Genetic background contributes to these differences. In our sleep analysis, to exclude the effects of genetic background on sleep, we out-crossed the flies to *w1118* (isoCS background) for at least 5 generations. However, there were still background differences between flies used in mutant analysis (Figure 1, Figure 2) and neural manipulation/neural rescue (Figure 5, Figure 6). wt flies had normal *white* gene (w+) at X chromosome. For mutant analysis in Figure 1 and Figure 2, to obtain the same genetic background as much as possible, after outcrossing, we replaced the X chromosome of the KO flies with that of wt. For neural manipulation/neural rescue in Figure 5, Figure 6, we didn’t replace X chromosome after outcrossing, so the X chromosome contains mutated *white* gene (*w1118* or w-). We showed the complete genotypes of flies used in each behavioral experiment as Table 1 in the revised manuscript.

In addition, between Figure 1EF and Figure 1—figure supplement 2J, the daytime sleep phenotype of Trh01 appears different. Between Figure 2, the rebound pattern appears very different in the same wild type flies. In 2C, the rebound mainly occurred during the daytime while in 2E, rebound occurred at night. What accounts for this difference?

Food used in the sleep analysis contributes to these differences. In Figure 1 and Figure 2, we used normal food (nutrition content shown in the Materials and methods) for sleep analysis, but in the 5HTP rescue experiments, to completely dissolve the 5HTP, we used 5% sucrose and 2% agar as food after eclosion. Previous studies have indicated diet and satiety could affect sleep and sleep homeostasis (e.g. Gatterson et.al. 2010 and Donlea et.al. 2011). We discussed this in the Discussion of the revised manuscript.

What was the time course of 5HTP treatment for experiments shown in Figure 2? The legend describes "Three day feeding of 2 mg/ml 5HTp" – did administration begin at the start of the baseline day and extend through the deprivation and recovery days? Or did it include 3 days prior to the baseline day? If the effects of 5HT on baseline sleep (Yuan et al., Curr Biol 16: 1051; 2006) were ramping up during the baseline day used to calculate sleep rebound, it would be impossible to dissociate the effects of 5HTP on baseline sleep and sleep rebound.

We have now described the 5HTP administration in details in the revised manuscript. Wt or Trh mutant flies were raised with normal food during larval stage and immediately transferred to 5HTP food (5% sucrose, 2% agar and 2mg/ml 5HTP) after eclosion. For immunostaining analysis, after three days feeding with 5HTP containing food, flies were dissected for immunohistochemistry. For behavioral analysis, flies were fed with 5HTP food after eclosion (usually 3-5 days), then transferred to glass tubes with 5HTP food for sleep analysis.

In the experiments shown in Figure 1—figure supplement 2, we observed that wt flies fed with 5HTP food increased about 30% sleep time in the whole day which indicated the role of 5HT in sleep promotion and is consistent with a previous study (Yuan et al., Curr Biol, year). In the sleep recovery experiment, we observed no significant difference between wt flies fed with mock and 5HTP, which indicated that 5HT was probably saturation for sleep recovery. Sleep duration is stable during the days we observed. Trh mutant flies fed with 5HTP increased sleep to the level a little higher that wt, and the daytime sleep was about 120 min. Although the daytime sleep of mutant flies with 5HTP is higher than wt, there could still be room for increase if the flies restore the recovery ability. Actually, the daytime sleep in wt flies can increase to >200 min after sleep deprivation. Recovery can be observed at the sleep-promotion condition and the effects of 5HTP on baseline sleep and sleep rebound can be dissociated.

Also, do Figure 2/2F analyze the same populations of animals? The data look to be the same for most genotypes, but data for Trh01+5HTP appear to be different (<30% sleep recovery in Figure 2 vs. >40% sleep recovery in Figure 2). If Figure 2 is from a single experiment, this should be stated along with population sizes (n).

Figure 2E/2F analyze the same populations of animals. We made mistakes in sample calculation of Trh mutant with 5HTP flies in the original Fig2F. We thank the reviewer for pointing out this. We have now corrected it in the revised manuscript.

Subsection “5-HT Regulation of Sleep Recovery After Deprivation Through its 2b Receptor”: Figure 2—figure supplement 1 should include the 24 hr time profile data after sleep deprivation since the quantification was performed using 24 hr data in Figure 2 and others.

The revised manuscript now includes the 24-hour time profile data after sleep deprivation.

Necessary genetic controls are missing for Figure 5 (Undriven UAS-Nachbach and UAS-Hid), 5E-F (Undriven LexAop-Flp, Tub>Gal80> & UAS-Hid) and 6B-E (5HT2b mutants with gal4 drivers but no UAS rescue).

In Figure 5, we have the controls of 5HT2b-LexA, 23E10-Gal4 and LexAop-Flp, Tub>Gal80>. We out-crossed each of the strains for 5 generations to reduce the genetic background differences, each of the strains showed stable sleep duration. As a remedy, we analyzed the controls from Figure 5—figure supplement 1, in which UAS-Nachbach, UAS-Hid were used as controls, the intersectional activation and ablation strain also showed significant sleep increase and decrease, respectively.

In Figure 5, the earlier version showed the wrong schematic illustration of analysis strategy in Figure 5, we have controls of 5HT2b, 23E10 and LexAop-Flp, Tub>Gal80>, UAS-Hid. We have corrected the schematic illustration in the revised manuscript.

In Figure 6, we have both positive (5HT2bGKO > UAS-5H2b that rescue in all 5HT2b neurons in 5HT2b mutant background, and UAS-5HT2b in wt) and negative controls (UAS-5HT2b in 5HT2b mutant background). The wt flies have *white* (w+) gene in X chromosome, which led to a different genetic background from other strains used in this experiment.

Figure 3 and Figure 3—figure supplement 1 appear to be nearly duplicate images (perhaps adjacent Z-sections). The authors should state this clearly so that readers do not get the impression that data are duplicated.

In the previous version, Figure 3 in Figure 3—figure supplement 1 were indeed the same images. Figure 3 is to show the expression pattern of Trh (we used strain Trh::Gal4 > UAS-mCD8GFP), and Figure 3—figure supplement 1 is to show expression comparison of Trh::Gal4 (Trh::Gal4 > UAS-mCD8GFP) and TrhGKO (TrhGKO > UAS-mCD8GFP). To avoid confusion, we are now using another image of Trh::Gal4 > UAS-mCD8GFP brain in the revised manuscript.

Subsection “Manipulation of 5HT2b and dFB intersectional neurons regulates sleep duration and homeostasis”, last paragraph: What was the efficiency of FLP-FRT recombination in the intersectional experiments in Figure 5?

We tested the efficiency of FLP-FRT recombination with immunostaining (23E10-Gal4, 5HT2b-LexA, UAS > stop> mCD8GFP, LexAop-Flp). For the 10 brains dissected, 4-8 neurons are labeled with 2-4 neurons on each side of the brain. The efficiency of FLP-FRT recombination is high enough for intersectional neural manipulations.

Subsection “Manipulation of 5HT2b and dFB intersectional neurons regulates sleep duration and homeostasis”, last paragraph: The description of either Figure 5 or 5E is wrong. The control genotype III shows impaired sleep rebound.

We showed the wrong schematic illustration of strategy in Figure 5 in our previous manuscript. We have corrected this in the revised manuscript.

Rescue manipulations in Figure 6 should probably have a wild-type (no transgene control), as the effects of overexpressing 5HT2B might not be apparent. Finally, some genotypes in Figure 6 are not fully clear. For animals positive for both "5HT2B mutant" and "5HT2B-GKO (genotype III), are these heterozygotes carrying two modified alleles at the 5HT2B locus?

As we had mentioned before, the wt control have white gene on X chromosome and the *w1118* flies are white eye. So we didn’t include the wt control in Figure 6. The other controls support our conclusions.

The genotypes in Figure 6 are not fully clear. We added the complete genotypes of flies used in each behavioral experiment as Table 1 in the revised manuscript.

Earlier work is not correctly cited in various parts of the manuscript (e.g. Becnel et al. 2011, Dierick et al. 2007 are cited as showing 5HT1a mutant flies have altered sleep, but these papers do not report these findings). In addition, the authors should discuss prior work pertinent to their study. For instance, the section on 5HTP feeding should refer to similar experiments done by Yuan et al. (2006) and discuss the possibility that increased sleep in the mutant background results from gain-of-function effects. In addition, findings with 5HT receptor and Trh GAL4 lines should be compared with those if earlier studies (Gnerer et al., 2015; Alekseyenko et al., 2010).

The revised manuscript has correct citations, new citations and discussions of previous work related to our current study.

The design and structure of alleles is not adequately described. Primer sequences, deletion junctions, et cetera should be described to enable readers to assess the nature of these alleles and validate them independently.

The revised manuscript has an additional table containing detail information for the design and structures of alleles used in the paper, including primer sequences for KI and KO (5’ arm and 3’ arm design), sgRNA information for CRISPR/CAS9 genome editing.

[Editors' note: further revisions were requested prior to acceptance, as described below.]

The manuscript has been improved but there are some remaining issues that need to be addressed before acceptance, as outlined below:1) Genetic controls that were requested in the previous decision letter were not included:"Necessary genetic controls are missing for Figure 5 (Undriven UAS-Nachbach and UAS-Hid), 5E-F (Undriven LexAop-Flp, Tub>Gal80> & UAS-Hid) & 6B-E (5HT2b mutants with gal4 drivers but no UAS rescue)."Undriven UAS-Nachbac/+ and UAS-Hid/+ controls are required for proper interpretation of 5B-C. Data for these lines are included in Figure 5—figure supplement 1 but baseline sleep levels in the other controls between these panels do not appear to be consistent. If the undriven UAS controls from Figure 5—figure supplement 1 are directly compared to data in Figure 5, it is not clear that genotype IV (the expression of Hid in dFB neurons expressing 5HT2b) has a significant decrease in daytime or whole day sleep from UAS-Hid controls or that the night-time sleep in genotype III (expression of NachBac in dFB neurons expressing 5HT2b) is higher than the UAS-Nachbac controls in Figure 5—figure supplement 1. These results require proper genetic controls to support the author's interpretations.

We thank the reviews for the comments. We repeated the experiments and added the necessary controls as requested. In Figure 5, we now have UAS-Nachbach, UAS-Hid, 23E10-Gal4/5HT2b-LexA and Tub>Gal80>/LexAop-Flp controls for activation and ablation of intersectional neurons.

In Figure 5, we now have UAS-Hid, 23E10-Gal4/5HT2b-LexA and Tub>Gal80>/LexAop-Flp for ablation of intersectional neurons. We show the changes in our revised manuscript.

Additionally, 23E10-GAL4 in a 5HT2b mutant background (with no UAS-5HT2b) is needed for Figure 6.Rescue manipulations in Figure 6 should probably have a wild-type (no transgene control), as the effects of overexpressing 5HT2B might not be apparent.

In Figure 6, we now have wt, UAS-5HT2b in 5HT2b mutant background and 23E10-Gal4 in mutant background as controls for rescue experiment. We have also rescued in all 5HT2b neurons as positive control. These experiments and RNAi manipulations further support that the 5HT2b gene in 23E10- and 5HT2b- intersectional neurons is necessary and sufficient in sleep regulation.

Finally, some genotypes in Figure 6 are not fully clear. For animals positive for both "5HT2B mutant" and "5HT2B-GKO (genotype III)", are these heterozygotes carrying two modified alleles at the 5HT2B locus?

We have now added the complete genotypes of the flies used. For genotype III (w-;UAS-5HT2b/+;5HT2b^GKO^/5HT2b^attp^), it is heterozygotes carrying two modified alleles at the 5HT2B locus, one allele is 5HT2b^GKO^(generated by Strategy II, replaced the gene with Gal4), the other is 5HT2b^attp^ (generated by Strategy I, replaced the gene with Gal4 and then deleted the insertion Gal4 with cre). Genotype III allows us to rescue 5HT2b expression in all 5HT2b neurons.

2) The interpretation of Figure 4 is not clear. These panels show that driving expression of 5HT2b::sfGFP in the dFB labels dFB neurons with GFP. If the authors want to show subcellular localization of 5HT2b in dFB neurons, the best strategy would be imaging 5HT2b::sfGFP protein-trap signal (see Figure 4) in flies that also contain R23E10-GAL4;UAS-RFP. That combination would permit clear colocalization with endogenous 5HT2b and dFB axons vs dendrites. Otherwise the authors should refrain from commenting on the subcellular location.

We are now clarifying the interpretation of Figure 4. We detected both strong anti-serotonin immunofluorescence and 5HT2b^sfGFP^ staining signal at dorsal FB region (Figure 4). There are two concerns. 1) There may be other 5HT2b positive neurons innervating to dFB, not only the ExFl2 neurons (23E10-Gal4 labeled dFB neurons) that promote sleep. 2) GPCRs are usually located at postsynaptic membrane. To determine whether the ExFl2 neurons contribute to the 5HT2bsfGFP staining signal at dorsal FB region, we crossed UAS-5HT2bsfGFP to 23E10-Gal4. We detected subcellular location of 5HT2bsfGFP in ExFl2 neurons. We found strong GFP in the presynaptic region of ExFl2 neurons, but not in the postsynaptic region nor cell body (Figure 4). These results suggest that 5HT2b receptor is located presynaptically in ExFl2 neurons. We have changed our interpretation of the Figure 4 in our revised manuscript.

3) The manuscript and title state that 5HT2b functions in a "single pair of neurons", though the intersectional genetic strategies applied to 5HT2b access perhaps two pairs of neurons. The "single pair of neurons" is from a different Gal4/LexA overlap experiment (Figure 4), which was not used for experiments involving 5HT2b. This is not a sticking point for the conclusion that 5HT2B functions in a subset of 23E10 neurons, but there is no need to overstate the result in the title and paper.

We have changed the statement of "single pair of neurons" into “a small subset” in both manuscript and title.

4) There continue to be a number of errors throughout the revised manuscript (text and figures), including several mentioned in the first round of review. These include mismatched citations to figures in the text (Results), and typographical mistakes. For instance, in the subsection “Genotypes in figures” under Figure 4—figure supplement 1, FRT-flanked stop alleles are misspelled as "Strop". There are also inconsistencies in the nomenclature.Note also:Figure 1 are still not legible.Figure 1 are out of order.Figure 1—figure supplement 4: in all three panels, legend on x-axis is missing for several genotypes.For Figure 2, the legend can be clarified to indicate that sleep gain is over 24h.

We thank the reviewers and editors for pointing out these. We have corrected the errors mentioned and revised and re-describe the nomenclature in the new manuscript.